# Sequential Best-Arm Identification with Application to P300 Speller

**Xin Zhou**                                                    *xinzhou@berkeley.edu*
*Division of Biostatistics, University of California at Berkeley*

**Botao Hao**                                                    *haobotao000@gmail.com*
*Google Deepmind*

**Tor Lattimore**                                                *tor.lattimore@gmail.com*
*Google DeepMind*

**Jian Kang**                                                    *jiankang@umich.edu*
*Department of Biostatistics, University of Michigan*

**Lexin Li**                                                     *lexinli@berkeley.edu*
*Division of Biostatistics, University of California at Berkeley*

**Reviewed on OpenReview:** *https: // openreview. net/ forum? id= QweNIIqvZf*

## Abstract

A brain-computer interface (BCI) is an advanced technology that facilitates direct communication between the human brain and a computer system, by enabling individuals to interact with devices using only their thoughts. The P300 speller is a primary type of BCI system, which allows users to spell words without using a physical keyboard, but instead by capturing and interpreting brain electroencephalogram (EEG) signals under different stimulus presentation paradigms. Traditional non-adaptive presentation paradigms, however, treat each word selection as an isolated event, resulting in a lengthy learning process. To enhance efficiency, we cast the problem as a sequence of best-arm identification tasks within the context of multi-armed bandits, where each task corresponds to the interaction between the user and the system for a single character or word. Leveraging large language models, we utilize the prior knowledge learned from previous tasks to inform and facilitate subsequent tasks. We propose a sequential top-two Thompson sampling algorithm under two scenarios: the fixed-confidence setting and the fixed-budget setting. We study the theoretical property of the proposed algorithm, and demonstrate its substantial empirical improvement through both simulations as well as the data generated from a P300 speller simulator that was built upon the real BCI experiments.

## 1 Introduction

A brain-computer interface (BCI) is a groundbreaking technology that enables direct communication between the brain and an external device or computer system. It involves the use of various sensors, such as electroencephalography (EEG), electrocorticography (ECoG), or implantable neural electrodes, which detect and record the electrical signals produced by the brain. The brain signals are then processed by machine learning algorithms to interpret and extract meaningful commands and intentions. BCI holds immense potential for a wide range of applications in medicine, rehabilitation, and human augmentation. For instance, it provides a valuable communication aid for individuals with disabilities (Wolpaw et al., 2018).

The P300 speller is a primary type of BCI system that allows users to select characters or spell words on a computer screen without using a physical keyboard but instead brain signals. It is based on the P300 event-

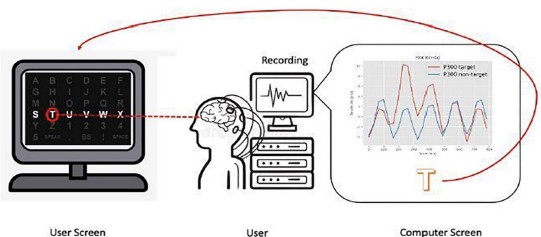

Figure 1: A schematic illustration: For each character or word the user intends to type, they focus their attention on the desired character or word on the virtual screen. To identify this character or word, the system presents a sequence of flashes on the virtual screen to the user. The user responds to these flashes, eliciting target or non-target brain signals. These EEG signals are captured by the electrodes, analyzed within a fixed time window after each flash, and transformed to a classifier score by a pre-trained classifier. A higher score indicates that the flash is more likely to contain the desired character or word. Consider the identification of a set of words the user intends to type as an example. It can be formulated as a series of best-arm identification problems. Here the system (not the user) acts as an agent, and it displays $J$ words on the virtual screen. A task means to identify the single word the user intends to type, which forms a single bandit environment. There are totally $M$ words the user intends to type, so $M$ total tasks. An action $a_j$, or say, pulling the $j$th arm, refers to flashing the $j$th word on the screen out of totally $J$ candidate words. At the $m$th task and the $t$th step, the agent selects an action $A_{t,m}$ from the action set $\mathcal{A} = \{a_1, \ldots, a_J\}$, and receives a reward $R_{t,m}$, which corresponds to the classifier score from the flash. The optimal arm $A_m^* \in \mathcal{A}$ means flashing the word the user intends to type in the $m$th task, and the decision $\psi_m$ refers to the word the system believes the user intends to type.

related potential (ERP), which is a brain response, in the form of a specific pattern of voltage fluctuation, that occurs approximately 300 milliseconds after a relevant target stimulus is presented. Specifically, the system presents an individual character or word through a sequence of flashes, with each flash being a stimulus, usually in a grid-like layout on a virtual screen for the user. If the flash contains the character or word the user wishes to type, an ERP is detected and recorded by a scalp EEG cap or a similar device. In that case, the user elicits a target brain signal. Otherwise, the user elicits a non-target brain signal. The recorded EEG signals are then analyzed by signal processing and machine learning algorithms, which detect the target stimulus and determine the target character. Figure 1 gives an illustration of the P300 speller.

A key limitation of the existing system is that the stimuli are usually presented in a fixed and predetermined fashion. In addition, when presenting a word (i.e., a collection of characters), or a sentence (i.e., a collection of words), the system treats each character or each word independently, and totally ignores the inherent relations among the characters or words. As a result, a large number of stimulus flashes are usually required to achieve a certain level of accuracy of character or word identification. A language model essentially defines a collection of conditional probability distributions over the next token given the past tokens. Recently, large language models (LLMs) such as GPT-3 (Brown et al., 2020) have achieved striking success in natural language processing, and can produce coherent and human-like text. In this article, we aim to utilize language models as the prior information to improve the sampling efficiency of the P300 BCI system through an adaptive stimulus design.

**Contributions**: Our contributions are three-fold.

- We introduce a novel sequential best-arm identification formulation for the P300 speller BCI application. We treat each word the user wishes to type as an optimal arm, and formulate stimulus presentation as an adaptive selection problem. We leverage language models as an informative prior, and aim to identify the target sequence of words as soon as possible in the fixed-confidence setting, or to make as fewer mistakes as possible given a fixed number of flashes in the fixed-budget setting.

- We propose a sequential top-two Thompson sampling (STTS) algorithm that utilizes the prior information in a coherent way. We derive the probability error bound in the fixed-budget setting that explicitly

quantifies the prior effect through the conditional entropy of the prior distribution of the optimal arms. We also study the probability error bound in the fixed-confidence setting.

- We conduct intensive experiments using both simulations and data generated from a P300 speller simulator that was built based on real BCI experiments (Ma et al., 2022). We consider two ways to generate the target sentence, or say, the set of words, that a user wishes to type. First, we use GPT-3 (Brown et al., 2020) to generate two sets of words given two prompts. Next, we choose a full sentence from a benchmark phrase set (MacKenzie & Soukoreff, 2003), and a full sentence from a recent news article (Wong, 2024). We use GPT-2 (Radford et al., 2019) to inform the prior distribution, and demonstrate the substantial improvement over several state-of-the-art baseline algorithms that do not utilize the prior information.

**Related work**: We first review the literature on multi-armed bandits, then the literature on P300 BCI.

The multi-armed bandit (MAB) problem (Robbins, 1952) is a classic scenario in decision theory and reinforcement learning that studies the problem of balancing exploration and exploitation. It involves an agent that seeks to optimize actions that maximize expected rewards. The agent must explore the action space sufficiently to acquire the knowledge needed to exploit the best action. Best-arm identification (BAI) is a variant of MAB, where the learner's objective is to identify the optimal arm, i.e., the arm with the highest expected reward, with a high accuracy. BAI is especially relevant to the P300 speller problem, where the goal is to swiftly and accurately identify target characters or words.

For learning a single task, Even-Dar et al. (2002) first introduced best-arm identification in the fixed-confidence setting, ensuring a specified confidence level in the correctness of the identification. Audibert et al. (2010) studied the fixed-budget setting, where the goal is to identify the best arm within a given number of trials or budget. In the BAI problem, we use sample complexity, the number of samples or pulls needed to correctly identify the best arm, to measure the theoretical properties of the algorithms. Kaufmann et al. (2016) investigated optimal sample complexity, and Russo (2016) proposed the top-two Thompson sampling as an effective anytime sampling rule that does not depend on the confidence parameter. Its theoretical properties were studied in Russo (2016); Qin et al. (2017); Shang et al. (2020); Qin & Russo (2022); Jourdan et al. (2022). However, most existing asymptotic analyses did not quantify the prior effect. An exception is Qin & Russo (2022), who showed the prior effect through entropy in the simple regret setting. Jun et al. (2016) explored scenarios where multiple pulls of the arms can be conducted simultaneously or in batches, rather than sequentially.

For learning multiple tasks sequentially, Boutilier et al. (2020); Simchowitz et al. (2021); Kveton et al. (2021); Azizi et al. (2022) studied meta-learning in the context of Bayesian bandits for cumulative and simple regret minimization. They assumed that an unknown instance prior is drawn from a known meta-prior. Then each task is sampled independently from this instance prior. In contrast, we assume the sequence of tasks is sampled from a joint prior distribution such that each task is *not independent* of each other. This necessitates the prior-dependent analysis that has only been studied in the regret minimization setting (Russo & Van Roy, 2016; Hao et al., 2023).

For P300 BCI studies, there have recently emerged a number of proposals for adaptive stimulus selection. Speier et al. (2011) used language models to weigh the output of stepwise linear discriminant analysis (LDA) for classification confidence. Park & Kim (2012) framed the problem as a a partially observable Markov decision process (POMDP). However, as the number of states grows, the belief states grow exponentially for POMDP, and the problem becomes intractable for a real-time system with a large search space. Koçanaoğulları et al. (2018) formulated the problem as query selection optimization and utilized language models to obtain the prior. However, they focused on a single task setting. Naively extending their solution to multiple tasks in effect requires that the prior information of the previous tasks must all be correct. If an incorrect selection is made in one task, then the subsequent prior information would be incorrect, which would in turn derail the subsequent stimulus selection. Besides, they did not establish any theoretical guarantee. Ma et al. (2021) used Beta-Bernoulli Thompson sampling for adaptive stimulus selection, but did not formulate the problem as best-arm identification, and did not incorporate any prior from language models. We refer to Heskebeck et al. (2022) for a review of the multi-armed bandits approaches in the BCI setting.

## 2  Sequential best-arm identification

### 2.1  Problem formulation

We formulate the problem as an agent sequentially interacting with $M$ bandit environments, with each interaction referred to as a task. Each environment, indexed by $m \in [M] = \{1, \ldots, M\}$, is characterized by a random vector $\theta_m \in \mathbb{R}^J$ with a prior distribution $\nu_m(\cdot)$, which will be discussed in detail in Section 2.2. The action set is $\mathcal{A} = \{a_1, \ldots, a_J\} \subseteq \mathbb{R}^J$, where $a_j$ is the standard basis vector with the $j$th element equal to one and the rest all zero. At each task $m$ and step $t$, the agent selects an action $A_{t,m} \in \mathcal{A}$, and receives a reward $R_{t,m} = \langle A_{t,m}, \theta_m \rangle + \eta_{t,m}$, where $\langle \cdot, \cdot \rangle$ is the vector inner product, and $(\eta_{t,m})$'s are a sequence of independent standard Gaussian variables. The optimal arm, $A_m^* = \mathrm{argmax}_{a \in \mathcal{A}} \theta_m^\top a$, is also a random variable. Let $(\Omega, \mathcal{F}, \mathbb{P})$ denote a measurable space, $\mathcal{H}_{t,m}$ the history of task $m$ up to step $t$, $\mathcal{D}_m = (\mathcal{H}_{\tau_1,1}, \ldots, \mathcal{H}_{\tau_{m-1},m-1})$, and $\mathbb{P}_{t,m}(\cdot) = \mathbb{P}(\cdot | \mathcal{H}_{t,m}, \mathcal{D}_m)$.

We consider two distinct scenarios.

In the fixed-confidence scenario (Even-Dar et al., 2002), the agent chooses a policy $\pi^m = (\pi_{t,m})_{t=1}^\infty$. The horizon is not fixed in advance, as the agent decides a stopping time $\tau_m$ adapted to filtration, $\mathbb{F}^m = (\mathcal{F}_{t,m})_{t=0}^\infty$, with $\mathcal{F}_{t,m} = \sigma(A_{1,m}, R_{1,m}, \ldots, A_{t,m}, R_{t,m})$, where $\sigma$ is the Borel $\sigma$-algebra. The agent makes a decision $\psi_m$ at the end of the task. For a given confidence level $\delta \in (0,1)$, the objective is to output a sequence of arms that are optimal for each task with probability at least $1 - \delta$ as soon as possible.

In the fixed-budget scenario (Bubeck et al., 2009), the agent is given a budget $n$ for each task, choose a policy $\pi^m = (\pi_{t,m})_{t=1}^n$, and makes a decision $\psi_m$ at the end of the task. The objective is to make the cumulative probability that $\psi_m$ is sub-optimal as small as possible.

In the context of a P300 speller experiment, take the identification of a set of words as an example. An agent refers to the system (not the user). A task means identifying a single word the user intends to type, which forms a single bandit environment. An action, or say, pulling an arm, means flashing one word on the virtual user screen, and the reward is the classifier score computed based on the captured EEG signals and a pre-trained binary classifier. A decision is the word the system believes the user intends to type. A budget is the total number of flashes the system is to make for a single task. In this setup, $M$ is the total number of words the user intends to type, and $J$ is the total number of flashes.

We also comment that we formulate our problem as a series of best-arm identification tasks within the context of multi-armed bandits, rather than reinforcement learning, because in our setting, there is no obvious definition of the state. If we define the underlying true word as a state, then the action we take would not affect the future state. Therefore, we feel the bandits formulation is more suitable for our problem.

### 2.2  Prior specification

A language model $\rho$ defines a collection of conditional probability distributions $(\rho_1, \ldots, \rho_M)$, where $\rho_m$ denotes a probability distribution over the $m$th word given the first $m - 1$ words, $m \in [M]$. In a P300 speller experiment, the word that an individual attempts to type is viewed as the optimal arm. The joint distribution over the collection of optimal arms $(A_1^*, \ldots, A_M^*)$ can be written through the chain rule:

$$\rho(A_1^*, \ldots, A_M^*) = \prod_{m=1}^M \rho_m = \prod_{m=1}^M \mathbb{P}\left(A_m^* = \cdot | A_1^*, \ldots, A_{m-1}^*\right). \tag{2.1}$$

For each task, the prior distribution of $\theta_m$ can be specified through

$$\nu_m = \mathbb{P}\left(\theta_m \in \cdot | A_1^*, \ldots, A_{m-1}^*\right) = \sum_{j=1}^J \Big\{ \underbrace{\mathbb{P}\left(\theta_m \in \cdot | A_m^* = a_j, A_1^*, \ldots, A_{m-1}^*\right)}_{\text{prior of the conditional mean reward}} \times \underbrace{\mathbb{P}\left(A_m^* = a_j | A_1^*, \ldots, A_{m-1}^*\right)}_{\text{prior of the optimal arm}} \Big\},$$

$$\tag{2.2}$$

which is a mixture distribution. While the prior of the optimal arm can be defined by (2.1), there are several ways to specify the prior of the conditional mean reward, depending on the problem setting. For the P300 speller experiment, the reward only differs upon whether the stimulus is a target or a non-target. Consequently, all sub-optimal arms share the same mean reward. It is thus natural to assume that,

$$\theta_m | A_m^* = a_j, A_1^*, \ldots, A_{m-1}^* \overset{d}{=} (\mu, \ldots, \underbrace{\mu + \Delta}_{j\text{th}}, \ldots, \mu),$$

where $\mu \sim \mathcal{N}(0, \sigma_0^2)$, and $\Delta \sim \exp(\sigma_1)$, for some $\sigma_0, \sigma_1 > 0$. We note that, for the P300 speller experiment, there are often offline data available so that $\Delta$ can be accurately estimated (Ma et al., 2021).

## 3  Sequential top-two Thompson sampling procedure

We propose a sequential top-two Thompson sampling (STTS) algorithm that utilizes the prior information in a coherent way. We also develop the corresponding stopping rule and the decision rule for both the fixed-confidence setting and the fixed-budget setting.

### 3.1  Sampling procedure

Our proposed STTS algorithm requires a posterior sampling oracle that can be obtained exactly when a conjugate prior is used, or through various approximation methods, such as Markov chain Monte Carlo.

**Definition 3.1** (Posterior sampling oracle). *Given a prior $\nu_m$ over $\theta_m$ and history $\mathcal{H}_{t,m}, \mathcal{D}_m$, the posterior sampling oracle, abbreviated as SAMP, is a subroutine that returns a sample from the posterior distribution $\mathbb{P}_{t,m}(\theta_m \in \cdot)$. Multiple calls to the procedure result in independent samples.*

Our STTS algorithm extends the top-two Thompson sampling algorithm of Qin & Russo (2022); Russo (2016) that was designed for a single task setting now to the sequential multiple tasks setting, where we sequentially call the language model $\rho$ to construct an informative prior. At task $m$ and step $t$, STTS first draws a posterior sample $\widetilde{\theta}_{t,m}$ using SAMP as well as the language model $\rho$, and sets $A_{t,m,1} = \text{argmax}_{a \in \mathcal{A}} \widetilde{\theta}_{t,m}^\top a$. Then STTS repeatedly samples from SAMP to obtain $\widetilde{\theta}_{t,m}$, and sets $A_{t,m,2} = \text{argmax}_{a \in \mathcal{A}} \widetilde{\theta}_{t,m}^\top a$ until $A_{t,m,1} \neq A_{t,m,2}$. We pick $A_{t,m} = A_{t,m,1}$ with probability equal to $\beta \in (0,1]$, and $A_{t,m} = A_{t,m,2}$ with probability equal to $1 - \beta$. In practice, we recommend $\beta = 1/2$, following Qin & Russo (2022). We also comment that SAMP is a subroutine that is frequently used in the literature (see, e.g., Russo et al., 2018; Jourdan et al., 2022). It can be obtained exactly when a conjugate prior is used, or approximately through various methods such as MCMC and variational inference.

### 3.2  Stopping rule and decision rule

For the fixed-confidence setting, we employ the Chernoff stopping rule introduced by Garivier & Kaufmann (2016); Shang et al. (2020). More specifically, let $N_{t,i}$ denote the number of pulls of arm $a_i$ before step $t$ for each task, $\mu_{t,i}$ the posterior mean for arm $a_i$, and $\mu_{t,i,j} = (N_{t,i}\mu_{t,i} + N_{t,j}\mu_{t,j})/(N_{t,i} + N_{t,j})$. We further define $Z_{t,i,j} = N_{t,i}\text{KL}(\mu_{t,i}, \mu_{t,i,j})$ for any two arms $a_i$ and $a_j$, and define $Z_t(a_i, a_j) = 0$ if $\mu_{t,j} \geq \mu_{t,i}$, and $Z_t(a_i, a_j) = Z_{t,i,j} + Z_{t,j,i}$ otherwise. Here we follow Garivier & Kaufmann (2016) and focus on the simple case where the distributions are fully parameterized by their means, and we use $\text{KL}(\mu_1, \mu_2)$ to denote the KL-divergence between the two distributions with means $\mu_1$ and $\mu_2$, respectively. For each task $m$, the Chernoff stopping rule is,

$$\tau_m = \inf \left\{ t \in \mathbb{N} : \max_{a_i \in \mathcal{A}} \min_{a_j \in \mathcal{A} \setminus \{a_i\}} Z_t(a_i, a_j) \geq \gamma_{t,\delta} \right\},$$

where $\gamma_{t,\delta}$ is the threshold parameter, and $\delta$ controls the level of confidence. As noted in Shang et al. (2020), $Z_t(a_i, a_j)$ can be interpreted as a generalized likelihood ratio statistic. As the agent interacts with $M$ tasks sequentially, we employ the Bonferroni correction to handle multiple comparisons (Dunn, 1961). As such, the family-wise error rate $\delta_M = \delta/M$, and the level of confidence for each task becomes $1 - \delta/M$. For the

---

**Algorithm 1** Sequential best-arm identification.

**Input:** The LLM $\rho$, the sampling oracle `SAMP`, the level of confidence $\delta$, the sampling parameter $\beta$.
**Output:** A sequence of recommended actions $\{\psi_1, \ldots, \psi_M\}$.
**for** $m \in [M]$ **do**
    Construct the informative prior based on $\rho$ and the history $\{\psi_1, \ldots, \psi_{m-1}\}$.
    Set $t = 0$.
    **while** TRUE **do**
        Set $t = t + 1$.
        Sample $\widetilde{\theta}_{t,m}$ from `SAMP`, and set $A_{t,m,1} = A_{t,m,2} = \mathrm{argmax}_{a \in \mathcal{A}} \widetilde{\theta}_{t,m}^\top a$.
        **while** $A_{t,m,1} = A_{t,m,2}$ **do**
            Re-sample $\widetilde{\theta}_{t,m}$ from `SAMP`, and set $A_{t,m,2} = \mathrm{argmax}_{a \in \mathcal{A}} \widetilde{\theta}_{t,m}^\top a$.
        **end while**
        Sample $C_{t,m} \sim \mathrm{Bernoulli}(\beta)$.
        **if** $C_{t,m} = 1$ **then**
            Set $A_{t,m} = A_{t,m,1}$.
        **else**
            Set $A_{t,m} = A_{t,m,2}$.
        **end if**
        Pull arm $A_{t,m}$, and receive $R_{t,m}$.
        Update `SAMP`, $N_{t,i}$, $\mu_{t,i}$, $\mu_{t,i,j}$.
        **if** $\max_{a_i \in \mathcal{A}} \min_{a_j \in \mathcal{A} \setminus \{a_i\}} Z_t(a_i, a_j) \geq \gamma_{t, \delta_M}$ **then**
            Identify the best arm $\psi_m$.
            STOP and break the while loop
        **end if**
    **end while**
**end for**

---

final decision rule, we choose the Bayes optimal decision rule, $\psi_m = \mathrm{argmax}_{a_i} \mu_{\tau_m, i}$. We summarize the full procedure for the fixed-confidence setting in Algorithm 1.

For the fixed-budget setting, the only change is that we stop the algorithm when some pre-specified budget constraint is met.

## 4 Theoretical properties

We first consider the fixed-budget setting. When the recommended word at the end of a task is wrong, we say the agent makes a mistake. The next theorem derives the corresponding probability error bound.

**Theorem 4.1.** *Consider a sequence of $M$ best-arm identification problems, and assume for each task $m$, all the sub-optimal arms have the same and known sub-optimality gap, $\Delta_m = \langle A_m^* - a, \theta_m \rangle$, for any $a \in \mathcal{A}$. If* `STTS` *is applied with $\beta \geq 1/2$ and with the Bayes optimal decision rule $\psi_m$, then for any positive integer-valued budget $n$,*

$$\frac{1}{M} \sum_{m=1}^{M} \mathbb{P}\left(\psi_m \neq A_m^*\right) \leq \sum_{m=1}^{M} \frac{\sqrt{h_m}}{\Delta_m} \frac{6}{M} \sqrt{\frac{\log\{J(1+n)\}}{1+n}} + \left(1 - \frac{1}{M} \sum_{m=1}^{M} \prod_{j=1}^{m-1} \left[1 - \frac{6}{\Delta_j} \sqrt{\frac{\log\{J(1+n)\} h_j}{1+n}}\right]\right),$$

$$(4.1)$$

*where $h_m = \mathbb{H}(A_m^* | A_{m-1}^*, \ldots, A_1^*)$, and $\mathbb{H}(\cdot | \cdot)$ is the conditional entropy.*

We discuss the individual terms in the error bound in (4.1). The first term in the bound is the main term, where $h_m = \mathbb{H}(A_m^* | A_{m-1}^*, \ldots, A_1^*)$ specifically characterizes the effect of the prior. In the P300 speller, the prior distribution of $A_m^*$ is informed by a language model such that $\mathbb{H}(A_m^* | A_{m-1}^*, \ldots, A_1^*)$ is much smaller than $\log(J)$. This can be validated through an experiment shown in Figure 2, left panel, where the entropy of the probability distribution over the next word outputted by GPT-2 is seen to be much smaller than an

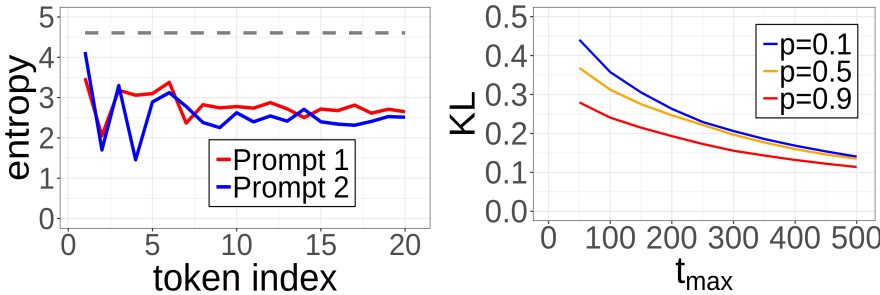

Figure 2: Left panel: the entropy of the next word under two different prompts (the red and blue curves) versus the entropy of a uniform distribution (the dashed line). Right panel: the KL divergence between the allocation rule of `STTS` and the optimal allocation rule versus the maximum number of steps when `STTS` stops. A larger $p$ indicates a stronger prior.

uniform distribution. The second term in the bound is the remainder term, which is the price the agent pays for making mistakes. Together, our bound in (4.1) explicitly characterizes the impact of the prior through the conditional entropy. When the prior is more informative, the conditional entropy decreases, and the error probability decreases. Moreover, the error probability converges to zero as the budget $n$ goes to infinity.

In this theorem, we have assumed the gaps for all sub-optimal arms are the same and known. This assumption allows us to use a simple regret guarantee to bound the error probability. We argue this is a reasonable assumption, especially for the application such as the P300 speller, since all the non-target stimuli have the same level of EEG responses and the gap can usually be estimated though the offline data (see, e.g., Ma et al., 2021; Kalika et al., 2017; Mainsah et al., 2017). On the other hand, when the gaps are unknown, how to establish the theoretical guarantee for the top-two TS under a fixed budget is still an open question even for the single task setting, and we leave it for future research.

We next consider the fixed-confidence setting. Choosing the threshold $\gamma_{t,\delta} = 4\ln(4 + \ln(t)) + 2C(\ln((J - 1)/\delta)/2)$ for some constant $C$, and applying Theorem 1 of Shang et al. (2020) leads to an asymptotic sample complexity bound for `STTS`, coupled with the Bonferroni correction, as,

$$\limsup_{\delta \to 0} \frac{\sum_{m=1}^{M} \mathbb{E}[\tau_m]}{\log(1/\delta)} \le \sum_{m=1}^{M} \int \frac{2J}{\Delta_m} \, \mathrm{d}\nu_m(\Delta_m) \,,$$

where the expectation is taken over the prior of $\theta_m$ on both sides. We remark that, this asymptotic result can not fully characterize the prior effect, as the asymptotic complexity measure only depends on the prior of the gap $\Delta_m$, rather than the prior of the position of the optimal arm. Typically, the overall sample complexity consists of two parts: the cost due to the optimal allocation rule, and the cost due to finding the optimal allocation rule. The latter is a lower-order term with respect to $\log(1/\delta)$. We conjecture that a more informative prior could greatly reduce the cost of finding the optimal allocation rule in the finite time. We conduct a numerical experiment to verify our conjecture. The detailed simulation setting is described in Appendix B. We report in Figure 2, right panel, the KL-divergence between the asymptotically optimal allocation rule and the allocation rule induced by `STTS`. It is clearly seen that, when the prior is stronger, as reflected by a larger value of the parameter $p$ that controls the strength of the prior, the allocation rule induced by `STTS` converges faster to the asymptotically optimal allocation rule. Meanwhile, a rigorous theoretical analysis requires a much more involved finite-time dependent analysis for top-two TS, and we leave it as future research.

## 5 Synthetic experiments

### 5.1 Simulation setup

We carry out simulations to investigate the empirical performance of our method and the effect of prior. We compare with the following baselines solutions.

- Vanilla top-two Thompson sampling (`VTTS`) (Russo, 2016; Qin & Russo, 2022): the top-two Thompson sampling that does not use any informative prior from the LLM.

- Random Policy (`Random`): the uniform sampling.

- Batch Racing (`BR`) (Jun et al., 2016): A frequentist algorithm that uses confidence interval to identify best arms in the fixed-confidence setting.

In our comparison, `VTTS` and `Random` use the same stopping rule and decision rule as described in Section 3.2, while `BR` is only for the fixed-confidence setting. There are several other popular best-arm identification algorithms, e.g., Jamieson & Nowak (2014); Garivier & Kaufmann (2016), in the literature, but none of them is designed for the sequential task setting.

We define the prior for the mean reward $\theta_m$ through (2.2), which requires the specification for the prior of the optimal arm, and the prior of the conditional mean reward. We assume the prior of the optimal arms satisfies the Markov property, in that the distribution of the $m$th optimal arm depends only on the $(m-1)$th optimal arm. More specifically, the optimal arm for $m$th task is sampled from $\mathbb{P}(A_m^* = a_j) = p$ if $j \in \{\text{supp}(A_{m-1}^*) + 1, \text{supp}(A_{m-1}^*) + 1 - J\} \cap [J]$, and $\mathbb{P}(A_m^* = a_j) = (1-p)/(J-1)$ if $j \in \{\text{supp}(A_{m-1}^*) + 1, \text{supp}(A_{m-1}^*) + 1 - J\}^c \cap [J]$, where $\text{supp}(v)$ denotes the position where the non-zero entry for a unit vector $v$ is located, $S^c$ denotes the complement of a set $S$, and $p \in [0,1]$ is a parameter that controls the strength of the prior, with a larger $p$ indicating a stronger prior.

Next, we set the prior of the mean reward of the $j$th arm to follow a Gaussian mixture distribution,

$$\mathbb{P}(\theta_{m,j} \in \cdot | A_1^*, \ldots, A_{m-1}^*) = p_{m,j}\mathcal{N}(\mu + \Delta, \sigma_0^2) + (1 - p_{m,j})\mathcal{N}(\mu, \sigma_0^2),$$

where we set the prior parameters as $\mu = 0, \sigma_0^2 = 0.2, \Delta = 2$. Moreover, when there is an oracle or external resource that reveals the identity of the optimal arm at the end of each task, `STTS` can start with an exact prior, which we call `STTS-Oracle`. We set $p_{m,j} \in [0,1]$ for different algorithms as follows.

- For `STTS`, we set $p_{m,j} = \mathbb{P}(A_m^* = a_j | A_{m-1}^* = \psi_{m-1})$ for $j \in [J]$, where $\{\psi_1, \ldots, \psi_{m-1}\}$ are the recommended actions.

- For `STTS-Oracle`, we set $p_{m,j} = \mathbb{P}(A_m^* = a_j | A_{m-1}^* = a_{m-1}^*)$ for $j \in [J]$, where $\{a_1^*, \ldots, a_{m-1}^*\}$ denote the realized instances of the optimal arms.

- For `VTTS`, we set $p_{m,1} = \ldots = p_{m,J} = J^{-1}$.

Moreover, let $\mu_{m,t,j}$ denote the posterior mean, and $\sigma_{m,t,j}^2$ the posterior variance.

We take one sentence as one experiment, and one word in the sentence as one task, with each sentence consisting of $M = 20$ words. We replicate each experiment $B = 200$ times. We vary the number of arms $J \in \{10, 20\}$, and set the confidence level $\delta = 0.1$.

We consider two accuracy measures for each experiment. One is the 0-1 accuracy, where as long as the agent makes at least one mistake among 20 words, we mark that experiment a failure. The other is the average accuracy, where we compute the percentage of correctly identified words among a sentence: $\sum_{m=1}^{M} \sum_{b=1}^{B} \mathbb{I}\{\psi_m^b = A_{m,b}^*\}/(BM)$, where $\psi_m^b, A_{m,b}^*$ denotes the recommended action and the optimal action for the $m$th task of the $b$th experiment, respectively.

We make a few comments. First, our proposed method does *not* require the Markov assumption. We have only adopted the Markov setting in our synthetic experiments for simplicity. Second, due to the Gaussian

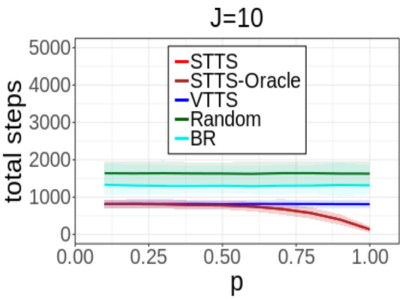 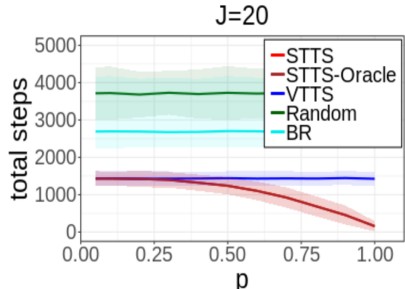

Figure 3: The total number of steps versus the prior strength $p$ for the fixed-confidence setting. A smaller number of total steps indicates a better performance of the method, and a larger $p$ indicates a stronger prior. The shaded area represents the 95% confidence interval.

reward noise, the posterior distribution retains the form of a Gaussian mixture distribution. As such, the posterior sampling oracle `SAMP` can be easily derived. Finally, we note that the proposed `STTS` is a fairly general algorithm that can be coupled with any prior. As an illustration, we consider a different prior specification in Appendix C.

## 5.2 Fixed-confidence setting

We first consider the fixed-confidence setting. We vary the parameter in the prior of the optimal arm $p \in \{J^{-1}\} \cup \{0.1, 0.2, \dots, 1.0\}$. A larger $p$ indicates a stronger prior effect, and $p = J^{-1}$ means a non-informative prior. In our setting, we stop the algorithm when the Chernoff stopping rule is satisfied; i.e.,

$$\tau_m = \min\left\{t : \min_{a_j \neq \psi_{t,m}} \frac{\mu_{m,t,\text{supp}(\psi_{t,m})} - \mu_{m,t,j}}{\sqrt{\sigma^2_{m,t,\text{supp}(\psi_{t,m})} + \sigma^2_{m,t,j}}} \geq \gamma_t\right\},$$

where $\psi_{t,m} = \operatorname{argmax}_a a^\top \mu_{m,t}$, and $\gamma_t = [2\log\{\log(t)M/\delta\}]^{1/2}$. Here, we approximate the KL-divergence of two Gaussian mixture distributions by the KL-divergence of two Gaussian distributions with the same mean and variance (Hershey & Olsen, 2007). Moreover, as the theoretical stopping rule of `BR` is conservative, we multiply its range by a factor of 0.25.

We compare different methods via the total number of steps under the varying prior strength $p$, whereas all methods achieve at least $1 - \delta = 0.9$ for the 0-1 accuracy. Figure 3 reports the results. We see that, with the prior becoming more informative, `STTS` clearly outperforms the baseline solutions `VTTS`, `BR`, and `Random`. The improvement also increases with an increasing number of arms. When the prior of the optimal arms is uniformly distributed, `STTS` coincides with `VTTS` as expected. Furthermore, `STTS` and `STTS-Oracle` exhibit similar performances. This is because, as the accuracy of `STTS` is consistently close to one, $\mathbb{P}(A^*_m = \cdot | A^*_{m-1} = a^*_{m-1})$ and $\mathbb{P}(A^*_m = \cdot | A^*_{m-1} = \psi_{m-1})$ are very close. We also briefly comment that, in real P300 speller experiments, it is impractical to run the number of steps into thousands. This step number can be adjusted by adjusting the accuracy measure. Here we mostly use this simulation example to illustrate the effect of the prior and the difference in performance of different solutions.

## 5.3 Fixed-budget setting

We next consider the fixed-budget setting. The sampling and decision rules remain the same, while the stopping rule is determined by whether the algorithm reaches a pre-specified budget of maximum step $t_{\max}$ per task. We vary $t_{\max} \in \{5, 10, 15, \dots, 100\}$, and $p \in \{J^{-1}, 0.5, 0.8, 1.0\}$.

We compare different methods via the average accuracy under the varying number of maximum step $t_{\max}$. Figure 4 reports the results. We see that, when the prior becomes more informative, `STTS` and `STTS-Oracle` reach a high accuracy with a smaller budget. When the prior is non-informative, `STTS` and `STTS-Oracle` perform similarly. We do not compare with `BR`, as it was designed for the fixed confidence setting only.

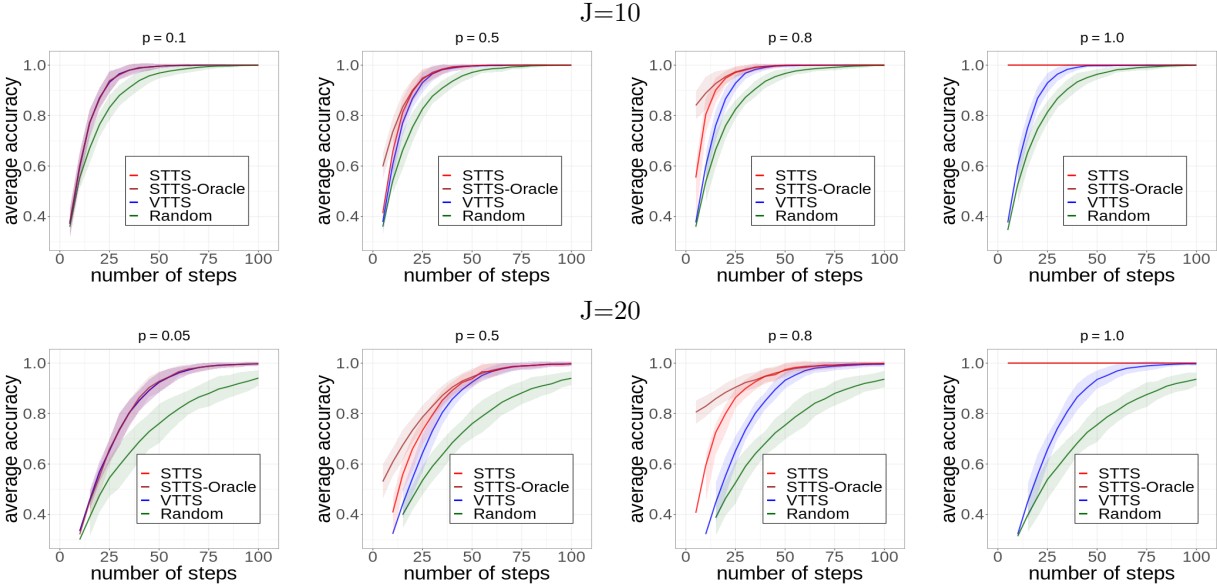

Figure 4: The average accuracy versus the maximum number of steps $t_{\max}$ under different prior strength $p$ for the fixed-budget setting. A larger $p$ indicates a stronger prior. The shaded area represents the 95% confidence interval.

# 6 P300 speller simulator experiment

We carry out an experiment by generating the brain EEG signal data using a realistic P300 speller simulator developed by Ma et al. (2022). We do not use any public BCI data. This is because our method employs an adaptive policy that influences both data generation and collection, in contrast to the random and non-adaptive baseline policy that is typically associated with such type of datasets. Thus, offline data alone is insufficient for our needs, and we are not aware of any online benchmark data that is suitable for our purpose. On the other hand, the simulator of Ma et al. (2022) has been built based upon real BCI experiments, and can adeptly generate signals that capture and match the complex spatial-temporal EEG patterns observed in real experiments. Moreover, we generate the sentence of words that a participant wishes to type using GPT-3 (Brown et al., 2020), and obtain the prior distribution using GPT-2 (Radford et al., 2019). This mimics the potentially imperfect prior information that commonly arises in real-world scenarios. All computations were done using CPUs on Google Colab Cluster.

More specifically, the P300 speller simulator of Ma et al. (2022) is built on a Bayesian generative model, and was trained with the data from actual BCI experiments. It employs novel split-and-merge Gaussian process priors to ensure an accurate emulation of the EEG signal generation mechanism inherent in the P300 speller in response to both target and non-target stimuli. For instance, the generated EEG signals exhibit the target ERPs of the frontal and central channels with a negative drop around 100 ms and an ascend to the first peak approximately at 250 ms latency. This pattern aligns seamlessly with the N100 and P300 patterns outlined in Stemmer & Whitaker (2008), underscoring the simulator's capacity to mirror real experimental conditions effectively. In our experiment, we set the number of electrodes to 16, the noise variance $\sigma_{\mathrm{EEG}}^2 \in \{1, 2.5\}$, the noise spatial correlation based on a Gaussian kernel function, the noise temporal correlation from an $AR(1)$ model with an autocorrelation 0.9, and the mean magnitude of the target stimulus five times that of the non-target stimulus. In accordance with the current practice (Manyakov et al., 2011), we first train a binary classifier for the P300 offline data based on stepwise linear discriminant analysis (Donchin et al., 2000; Krusienski et al., 2008). This classifier converts the raw EEG signals into the classifier scores, and we take these scores as the rewards in our setting. A higher score indicates that the EEG signal is more likely to correspond to a target stimulus.

Table 1: The total number of steps, i.e., the stimulus flashes, for the fixed-confidence setting using the P300 speller simulator and the GPT-generated set of words. Two prompts are used. A larger $\sigma^2_{\text{EEG}}$ indicates a larger noise level in the simulated EEG signals.

| Prompt | $\sigma^2_{\text{EEG}}$ | Method | total steps (std) | Prompt | $\sigma^2_{\text{EEG}}$ | Method | total steps (std) |
|--------|--------|--------|-------------------|--------|--------|--------|-------------------|
| 1 | 1 | STTS | 668.30 (136.5) | 2 | 1 | STTS | 824.80 (171.3) |
| | | VTTS | 1445.0 (237.2) | | | VTTS | 1441.1 (221.8) |
| | | Random | 6865.5 (826.6) | | | Random | 7016.6 (956.8) |
| | | BR | 2611.8 (112.6) | | | BR | 2662.0 (140.3) |
| | | BBTS | 1430.4 (203.5) | | | BBTS | 1421.7 (197.2) |
| | 2.5 | STTS | 1623.0 (240.6) | | 2.5 | STTS | 1877.4 (258.2) |
| | | VTTS | 2493.2 (275.5) | | | VTTS | 2445.8 (252.3) |
| | | Random | 13523.2 (829.9) | | | Random | 13517.0 (952.6) |
| | | BR | 4396.4 (322.8) | | | BR | 4441.7 (271.2) |
| | | BBTS | 1712.5 (248.1) | | | BBTS | 1694.0 (266.8) |

We consider two ways to generate the target sentence, or say, the set of words, that a user wishes to type. First, we use GPT-3 (Brown et al., 2020) to generate $M = 20$ words under two given prompts, *"The most popular food in the United States is"* as Prompt 1, and *"My favorite sport is"* as Prompt 2. Next, we choose a sentence with $M = 9$ words from a benchmark phrase set (MacKenzie & Soukoreff, 2003), and a sentence with $M = 15$ words and punctuation marks from a recent news article on *Nature* (Wong, 2024). We use GPT-2 (Radford et al., 2019) to inform the prior distribution and specify the prior of the optimal arms. That is, given an input token, GPT-2 specifies a probability distribution of the next token over the full vocabulary space. Besides, for the very first word in the set of words, we provide the prompt to GPT-2 to obtain the prior in the GPT-3 generation case, provide no information at all to GPT-2 in the case of using the sentence from the benchmark phrase set, and provide the preceding sentence to GPT-2 in the case of using the sentence for the news article. We truncate the vocabulary size of GPT-2 from the original size of 50257 to 100, following the top-$K$ sampling (Fan et al., 2018) and the nucleus sampling (Holtzman et al., 2019). This keeps the candidate words with the top-100 highest probabilities, and in effect substantially reduces the size of the action space to $J = 100$. We repeat each prompt $B = 100$ times, and set the confidence level at $1 - \delta = 0.9$. We report the results based on GPT-3 generation in this section, and report the results based on the chosen sentences in Appendix E.

We compare the algorithms studied in Section 5.1. When a wrong action is recommended, we let the experiment continue to run and reveal the identity of the optimal arm. As such, STTS and STTS-Oracle yield the same results. We also add the comparison with Beta-Bernoulli Thompson sampling (BBTS) algorithm of Ma et al. (2021).

For the fixed-confidence setting, we stop the algorithm when the Chernoff stopping rule is met. Table 1 reports the total number of steps, i.e., the stimulus flashes, whereas the 0-1 accuracy and the average accuracy for all methods are above 0.9. We see that, facilitated by the LLM-informed prior, STTS manages to reduce the total number of stimulus flashes by 25% to 50%, while maintaining about the same accuracy.

For the fixed-budget setting, we stop the algorithm when it reaches the pre-specified maximum number of flashes $t_{\max} \in \{5, 10, 15, \ldots, 150\}$. Figure 5 reports both the 0-1 accuracy and the average accuracy. We see that STTS achieves the highest accuracy compared to the alternative solutions. We also note that, the baseline random policy, which is currently used in most P300 experiments, requires about 150 steps to reach the 95% accuracy. This agrees with the practical observations that usually 100 to 150 steps are needed (Kindermans et al., 2012). In contrast, our method only requires 60 steps, which is a significant improvement.

# 7 Conclusion

The BCI technology has the potential to revolutionize human communication by bypassing physical constraints and interacting directly with the brain. In this article, we have proposed a sequential best-arm

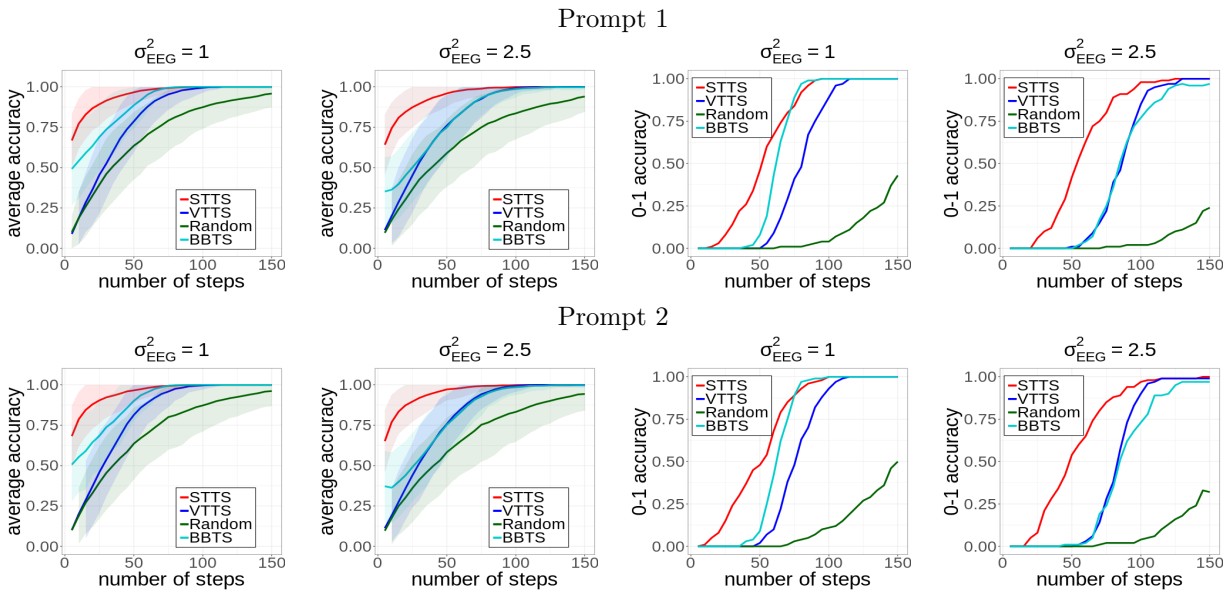

Figure 5: The average accuracy and the 0-1 accuracy for the fixed-budget setting using the P300 speller simulator and the GPT-generated set of words. Two prompts are used. A larger $\sigma^2_{\text{EEG}}$ indicates a larger noise level in the simulated EEG signals.

identification formulation for the P300 speller BCI system that greatly enhances its sampling efficiency. It integrates the prior information derived from LLMs with the top-two Thompson sampling algorithm. However, it is more than merely a simple extension, and offers several useful contributions. Notably, our algorithm extends Qin & Russo (2022); Koçanaoğulları et al. (2018) from the setting of a single task to that of multiple and dependent tasks, while relaxing the stringent requirement that all priors have to be correctly specified. Our theory, to the best of our knowledge, is the first work that establishes an explicit error probability bound for top-two Thompson sampling within a fixed-budget context. Finally, we apply our method to the P300 speller type BCI experiments, which we hope could pave the way for more adoption of adaptive bandits algorithms in this important application area.

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

## A   Proof of Theorem 4.1

We first consider the case when the recommended action is wrong, the subsequent task would use the wrong information. Suppose the prior is as specified in Section 2.2. Define the sub-optimality gap $\Delta_m = \langle A_m^*, \theta_m \rangle - \langle a, \theta_m \rangle$ for any $a \in \mathcal{A}, a \neq A_m^*$. Note that

$$\mathbb{E}\left[\langle A_m^*, \theta_m \rangle - \langle \psi_m, \theta_m \rangle\right] = \mathbb{E}\left[(\langle A_m^*, \theta_m \rangle - \langle \psi_m, \theta_m \rangle)\, \mathbb{I}\,(\psi_m \neq A_m^*)\right] + \mathbb{E}\left[(\langle A_m^*, \theta_m \rangle - \langle \psi_m, \theta_m \rangle)\, \mathbb{I}\,(\psi_m = A_m^*)\right]$$
$$= \mathbb{E}\left[\Delta_m \mathbb{I}\,(\psi_m \neq A_m^*)\right],$$

where $\psi_m$ is the recommended action at the end of task $m$. Since we assume all the sub-optimal arms have the same mean reward, and the sub-optimality gap is known, the error probability can be bounded through the simple regret:

$$\mathbb{P}\,(\psi_m \neq A_m^*) = \frac{1}{\Delta_m}\mathbb{E}\left[\langle A_m^*, \theta_m \rangle - \langle \psi_m, \theta_m \rangle\right],$$

where the expectation in the left-hand-side is with respect to the prior distribution of $\theta_m$. We then employ Proposition 1 of Qin & Russo (2022) for the top-two Thompson sampling for a single task. Define an event $\mathcal{E}_m = \{\psi_1 = A_1^*, \ldots, \psi_{m-1} = A_{m-1}^*\}$. As Qin & Russo (2022) requires the top-two Thompson sampling starting with a correct prior, we decompose the simple regret term based on $\mathcal{E}_m$ as,

$$\mathbb{E}\left[\langle A_m^*, \theta_m \rangle - \langle \psi_m, \theta_m \rangle\right] = \underbrace{\mathbb{E}\left[(\langle A_m^*, \theta_m \rangle - \langle \psi_m, \theta_m \rangle)\, \mathbb{I}(\mathcal{E}_m)\right]}_{I_1} + \underbrace{\mathbb{E}\left[(\langle A_m^*, \theta_m \rangle - \langle \psi_m, \theta_m \rangle)\, \mathbb{I}(\mathcal{E}_m^c)\right]}_{I_2}. \quad \text{(A.1)}$$

Applying Proposition 1 of Qin & Russo (2022), we obtain that,

$$I_1 \leq 6\sqrt{\log(J(1+n))\mathbb{H}(A_m^*|A_{m-1}^*, \ldots, A_1^*)(1+n)^{-1}}, \quad \text{(A.2)}$$

where $\mathbb{H}(\cdot|\cdot)$ is the conditional entropy. To bound $I_2$, we have that,

$$I_2 = \mathbb{E}\left[(\langle A_m^*, \theta_m \rangle - \langle \psi_m, \theta_m \rangle)\, \mathbb{I}(\mathcal{E}_m^c)\right]$$
$$\leq \Delta_m \left(1 - \mathbb{E}\left[\mathbb{P}\left(\psi_1 = A_1^*, \ldots, \psi_{m-1} = A_{m-1}^*\right)\right]\right)$$
$$= \Delta_m \left(1 - \mathbb{E}\left[\prod_{j=1}^{m-1} \mathbb{P}\left(\psi_j = A_j^*|\psi_{j-1} = A_{j-1}^*, \ldots, \psi_1 = A_1^*\right)\right]\right)$$
$$= \Delta_m \left(1 - \mathbb{E}\left[\prod_{j=1}^{m-1} \left(1 - \mathbb{P}\left(\psi_j \neq A_j^*|\psi_{j-1} = A_{j-1}^*, \ldots, \psi_1 = A_1^*\right)\right)\right]\right).$$

Applying Proposition 1 of Qin & Russo (2022) again, we have that,

$$\mathbb{P}\left(\psi_j \neq A_j^*|\psi_{j-1} = A_{j-1}^*, \ldots, \psi_1 = A_1^*\right) \leq \frac{6}{\Delta_j}\sqrt{\frac{\log(J(1+n))\mathbb{H}(A_j^*|A_{j-1}^*, \ldots, A_1^*)}{1+n}}. \quad \text{(A.3)}$$

Denote $p_m = 6\{\log(J(1+n))\mathbb{H}(A_m^*|A_{m-1}^*,\ldots,A_1^*)(1+n)^{-1}\}^{1/2}/\Delta_m$. Putting (A.1)-(A.3) together, we obtain that,

$$\mathbb{P}\left(\psi_m \neq A_m^*\right) \leq p_m + \left(1 - \prod_{j=1}^{m-1}(1-p_j)\right).$$

Summing over all the tasks, we obtain that,

$$\sum_{m=1}^{M}\mathbb{P}\left(\psi_m \neq A_m^*\right) \leq \sum_{m=1}^{M}p_m + \sum_{m=1}^{M}\left(1 - \prod_{j=1}^{m-1}(1-p_j)\right),$$

where the second term is the price the agent pays for the wrong prediction.

This completes the proof.

## B    Additional discussion on the theory

First, we note that, for the fixed-budget setting, there is another practical scenario where at the end of task $m$, the agent is given the identity of the optimal arm $A_m^*$. If the recommended action is wrong, the agent pays an extra price $c$. For P300 speller, if the system outputs a wrong recommendation, the participant will gaze at the backspace and the system will repeat the process until the right word is recommended. In this case, the event $\mathcal{E}_m = \{\psi_1 = A_1^*, \ldots, \psi_{m-1} = A_{m-1}^*\}$ always holds. Based on (A.2), the error probability can be bounded by

$$\frac{1}{M}\sum_{m=1}^{M}\mathbb{P}\left(\psi_m \neq A_m^*\right) \leq \sum_{m=1}^{M}\frac{\sqrt{h_m}}{\Delta_m}\frac{6}{M}\sqrt{\frac{\log(J(1+n))}{1+n}},$$

and the number of mistakes is bounded by $c\sum_{m=1}^{M}\mathbb{P}(\psi_m \neq A_m^*)$.

Next, we verify the role of prior in the fixed-confidence setting in our conjecture in Section 4. Specifically, we study how fast the allocation rule of each arm using STTS converges to the optimal allocation rule and how the information of the prior affects the convergence. Following the prior specified in Section 5 and the computation in Shang et al. (2020), the optimal allocation rule $p^*$ pulls the optimal arm with probability $\beta = 1/2$, and all the other arms with probability $1/(2J-2)$. We set the prior parameters $\mu = 0, \sigma_0^2 = 1, \Delta = 0.5$, the number of arms $J = 10$, and we vary $p \in \{0.1, 0.5, 0.9\}$. Define $p_{t,a_i} = N_{t,a_i}/t$ as the proportion of selections of arm $a_i$ before step $t$. The algorithm stops when it reaches the pre-specified maximum number of steps $t_{\max} \in \{50, 100, 150, \ldots, 500\}$ per task. Figure 2, right panel, reports the KL divergence $\text{KL}(p_t, p^*)$ that compares the STTS allocation rule and the optimal allocation rule. We see clearly that, when the prior becomes stronger, the allocation rule induced by STTS converges faster to the optimal allocation rule.

## C    Alternative prior specification

Our proposed algorithm STTS is general and can be coupled with any prior specification. As an illustration, we consider a Gaussian prior specification. Recall the language model $\rho_m$ only requires the previous best arm $A_{m-1}^*$ to define the prior distribution as $\nu_m = \mathcal{N}(\rho_m(A_{m-1}^*), \sigma_0^2 I)$, where $\sigma_0$ is the standard deviation parameter, and $\rho_m$ is specified as follows. We first introduce three matrices $U^{(1)}$, $U^{(2)}$, and $U^{(3)}$.

- For $U^{(1)}$, we set

$$U_{j_1,j_2}^{(1)} = \left\{\begin{array}{ll} 1, & (j_2 - j_1) \in \{1-J, 1\} \\ 0, & \text{otherwise} \end{array}\right.,$$

which means that the optimal arm in the next task is more likely to be the one immediately following the current optimal arm.

- For $U^{(2)}$, we set

$$U^{(2)}_{j_1,j_2} = \begin{cases} 1, & (j_1, j_2) \in \{(1,2),(2,3),\ldots,(J-2,1),(J-1,J),(J,J-1)\} \\ 0, & \text{otherwise} \end{cases},$$

  which consists of two groups where the first $(J-2)$ arms are in the first group, and the last two arms are in the second group.

- For $U^{(3)}$, we set

$$U^{(3)}_{j_1,j_2} = \begin{cases} 1, & (j_2 - j_1) \in \{1 - J, 1\} \\ 0.5, & (j_2 - j_1) \in \{2 - J, 2\} \\ 0, & \text{otherwise} \end{cases},$$

  which possesses more uncertainty than $U^{(1)}$.

Let $e_j$ be the unit vector with length $J$ with the $j$th entry being 1. We define $\rho_m(A^*_{m-1}) = \mu_0 (A^*_{m-1})^T U$, for $m \geq 2$, where $U$ takes one of the forms of $U^{(1)}$, $U^{(2)}$, and $U^{(3)}$, $\mu_0$ controls the magnitude of the best arm. For $m = 1$, we assign each arm to be the optimal arm with equal probability, and $\nu_1 = N(\mu_0 e_j, \sigma_0^2 I_J)$ if arm $j$ is the optimal arm. The instance $\theta_m$ is sampled from the prior distribution $\nu_m$. In this case, we set the posterior mean $\mu_{m,t,j} = \mathbb{E}(\theta_{m,j} \in \cdot | \mathcal{H}_{t,m} \mathcal{D}_m)$, and the posterior variance $\sigma^2_{m,t,j} = \text{Var}(\theta_{m,j} \in \cdot | \mathcal{H}_{t,m} \mathcal{D}_m)$. Moreover, we set the number of tasks $M = 20$, and the agent interacts with $M$ bandit instances with the confidence level $1 - \delta = 0.9$, and the reward noise variance $\sigma^2 = 1$. For VTTS, we specify the non-informative Gaussian prior as $\mu_{0,m,j} = 0$, and $\Sigma_{0,m,j} = 10^2$ for $j \in [J]$.

Table 2: The accuracy and the number of total steps with a Gaussian prior specification. A larger $\mu_0$ and a smaller $\sigma_0$ means a stronger prior information.

| $J$ | $\mu_0$ | $\sigma_0$ | Method | $U^{(1)}$ | | $U^{(2)}$ | | $U^{(3)}$ | |
|---|---|---|---|---|---|---|---|---|---|
| | | | | accuracy | steps | accuracy | steps | accuracy | steps |
| 5 | 4 | 0.5 | STTS | 100.0 | 60.0 | 100.0 | 50.8 | 98.5 | 307.5 |
| | | | VTTS | 100.0 | 253.1 | 100.0 | 251.4 | 99.5 | 715.9 |
| | | | Random | 100.0 | 405.1 | 100.0 | 406.1 | 98.0 | 1351.3 |
| | 5 | 0.5 | STTS | 100.0 | 47.1 | 100.0 | 47.3 | 99.5 | 76.6 |
| | | | VTTS | 100.0 | 184.1 | 100.0 | 183.3 | 99.5 | 370.6 |
| | | | Random | 100.0 | 295.7 | 100.0 | 296.6 | 99.5 | 708.5 |
| | 4 | 1 | STTS | 98.5 | 412.1 | 97.5 | 429.8 | 89.5 | 2008.0 |
| | | | VTTS | 98.5 | 644.3 | 98.5 | 655.8 | 89.0 | 2183.9 |
| | | | Random | 96.5 | 1052.1 | 98.5 | 1048.1 | 82.5 | 3332.3 |
| | 5 | 1 | STTS | 99.5 | 81.8 | 98.5 | 80.3 | 94.5 | 1115.3 |
| | | | VTTS | 99.5 | 267.6 | 99.5 | 273.2 | 93.5 | 1286.8 |
| | | | Random | 99.5 | 449.0 | 100.0 | 459.3 | 94.0 | 2088.8 |
| 10 | 4 | 0.5 | STTS | 100.0 | 62.6 | 98.5 | 62.2 | 98.5 | 455.5 |
| | | | VTTS | 100.0 | 480.9 | 100.0 | 482.1 | 99.5 | 1110.1 |
| | | | Random | 100.0 | 1030.0 | 99.5 | 1036.2 | 98.0 | 3173.1 |
| | 5 | 0.5 | STTS | 100.0 | 54.9 | 100.0 | 54.8 | 100.0 | 94.8 |
| | | | VTTS | 100.0 | 333.1 | 100.0 | 336.5 | 100.0 | 525.7 |
| | | | Random | 100.0 | 764.5 | 100.0 | 760.8 | 100.0 | 1517.3 |
| | 4 | 1 | STTS | 98.5 | 1588.8 | 96.0 | 1425.2 | 93.0 | 5362.4 |
| | | | VTTS | 99.0 | 2162.3 | 98.0 | 2145.4 | 94.5 | 5700.2 |
| | | | Random | 95.5 | 4803.8 | 95.0 | 4786.2 | 88.0 | 11594.4 |
| | 5 | 1 | STTS | 98.5 | 109.9 | 100.0 | 138.0 | 97.0 | 2341.1 |
| | | | VTTS | 100.0 | 597.4 | 100.0 | 616.5 | 97.5 | 2771.5 |
| | | | Random | 100.0 | 1375.8 | 98.5 | 1424.1 | 96.5 | 6288.7 |

We repeat the experiments 200 times for $J \in \{5, 10\}$, $\mu_0 \in \{4, 5\}$, and $\sigma_0 \in \{0.5, 1\}$. Table 2 reports the accuracy and the number of total steps. It is seen from the table that `STTS` achieves the smallest number of total steps, while maintaining the desired accuracy. When $\mu_0$ is large and $\sigma_0$ is small, which indicates a strong prior information, the improvement of `STTS` over `VTTS` in terms of the total steps is more evident.

## D   Computational complexity evaluation

In the P300 speller system, a sequence of flashes is presented on a virtual screen to the user. The time interval between two consecutive flashes is a constant, denoted as $c_0$, which is also the time taken for each step. We follow Ma et al. (2021), and set this time interval $c_0 = 160$ ms. We observe that our proposed algorithm as well as the alternative algorithms require much shorter time than $c_0$ in each step. As such, instead of reporting the computational time alone, we employ the information transfer rate (ITR) (Wolpaw et al., 1998), and the BCI utility (Utility) (Dal Seno et al., 2009), as the metrics to evaluate the overall computational complexity of all the algorithms. Specifically, we compute the two measures as,

$$
\begin{aligned}
\text{ITR} &= \frac{1}{c_0} \left\{ \log_2 J + \phi \log_2 \phi + (1 - \phi) \log_2 \left( \frac{1 - \phi}{J - 1} \right) \right\}, \\
\text{Utility} &= \frac{1}{c_0} \max \left\{ 0, (2\phi - 1) \log_2(J - 1) \right\},
\end{aligned}
$$

which take into account the time interval $c_0$, the average accuracy $\phi$, and the total number of arms $J$.

Figure 6 reports the ITR and Utility measures for the P300 speller simulator example studied in Section 6. We see that `STTS` achieves the best performance in terms of the ITR and the utility measure.

## E   Additional P300 simulator experiment

In Section 6, we have generated the target set of words that a user wishes to type using GPT-3. In this section, we consider another scenario of using some chosen sentences. Specifically, we choose Sentence 1,

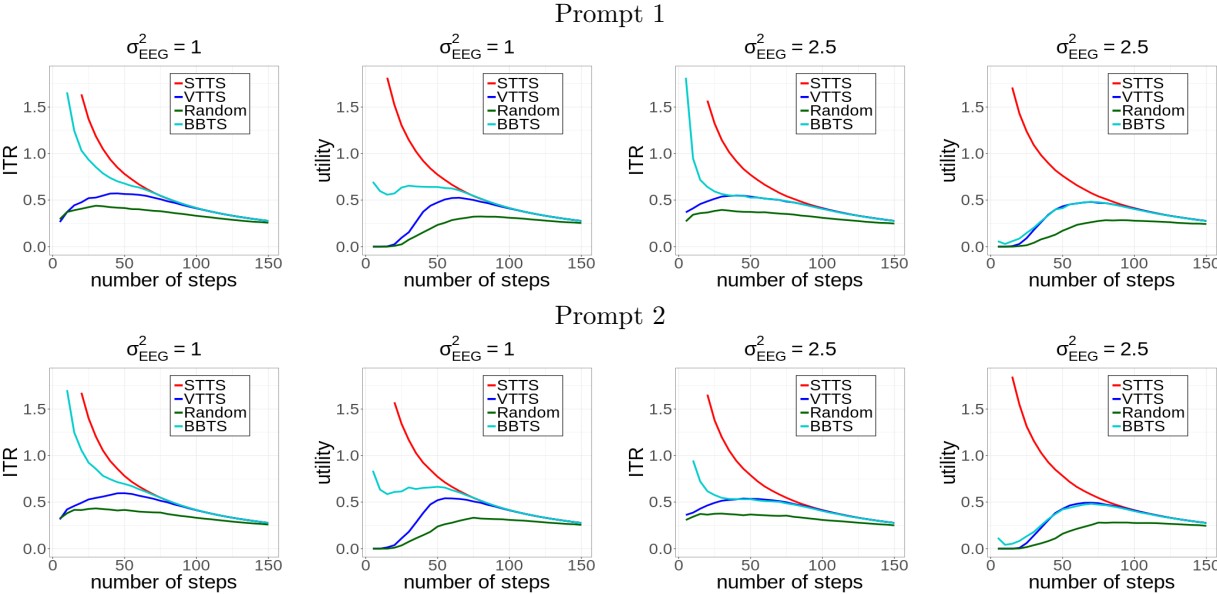

Figure 6: The information transfer rate (ITR) and the utility measure for the fixed-budget setting using the P300 speller simulator and the GPT-generated set of words. Two prompts are used. A larger $\sigma_{\text{EEG}}^2$ indicates a larger noise level in the simulated EEG signals.

Table 3: The total number of steps, i.e., the stimulus flashes, for the fixed-confidence setting using the P300 speller simulator and the chosen sentences. Two sentences are used, one from a benchmark phrase set, and the other from a news article. A larger $\sigma^2_{\text{EEG}}$ indicates a larger noise level in the simulated EEG signals.

| Sentence | $\sigma^2_{\text{EEG}}$ | Method | total steps (std) | Sentence | $\sigma^2_{\text{EEG}}$ | Method | total steps (std) |
|---|---|---|---|---|---|---|---|
| 1 | 1 | STTS | 368.71 (69.66) | 2 | 1 | STTS | 443.05 (86.39) |
| | | VTTS | 623.27 (132.1) | | | VTTS | 1081.42 (190.8) |
| | | Random | 3071.16 (609.9) | | | Random | 5055.45 (711.3) |
| | | BR | 1148.36 (77.54) | | | BR | 1918.71 (99.80) |
| | | BBTS | 623.81 (112.8) | | | BBTS | 1006.45 (154.4) |
| | 2.5 | STTS | 596.58 (101.9) | | 2.5 | STTS | 830.07 (116.7) |
| | | VTTS | 872.5 (136.1) | | | VTTS | 1668.23 (193.1) |
| | | Random | 5750.19 (729.8) | | | Random | 9930.46 (809.5) |
| | | BR | 1879.16 (174.9) | | | BR | 3207.29 (220.9) |
| | | BBTS | 749.81 (143.9) | | | BBTS | 1286.29 (208.2) |

*"Rent is paid at the beginning of the month"* with $M = 9$ words, from a benchmark phrase set (MacKenzie & Soukoreff, 2003). We choose Sentence 2, *"It substantially raises the risk of diseases including cancer, heart disease and diabetes."* with $M = 15$ words and punctuation marks, from a recent news article (Wong, 2024). Note that this news article was published in April, 2024, and thus is not in the GPT-2's training data which is up to October, 2019 only. Similarly as before, we use GPT-2 (Radford et al., 2019) to obtain the prior distribution. However, for the very first word of a sentence, we provide no information at all to GPT-2 for Sentence 1, and we provide the immediately preceding sentence, *"The health harms of smoking tobacco have been established for decades -"* to GPT-2 for Sentence 2. We again conduct experiments for both the fixed-confidence setting and the fixed-budget setting.

For the fixed-confidence setting, Table 3 reports the total number of steps, or flashes, whereas the 0-1 accuracy and the average accuracy for all methods are above 0.9. We again see that, facilitated by the LLM-

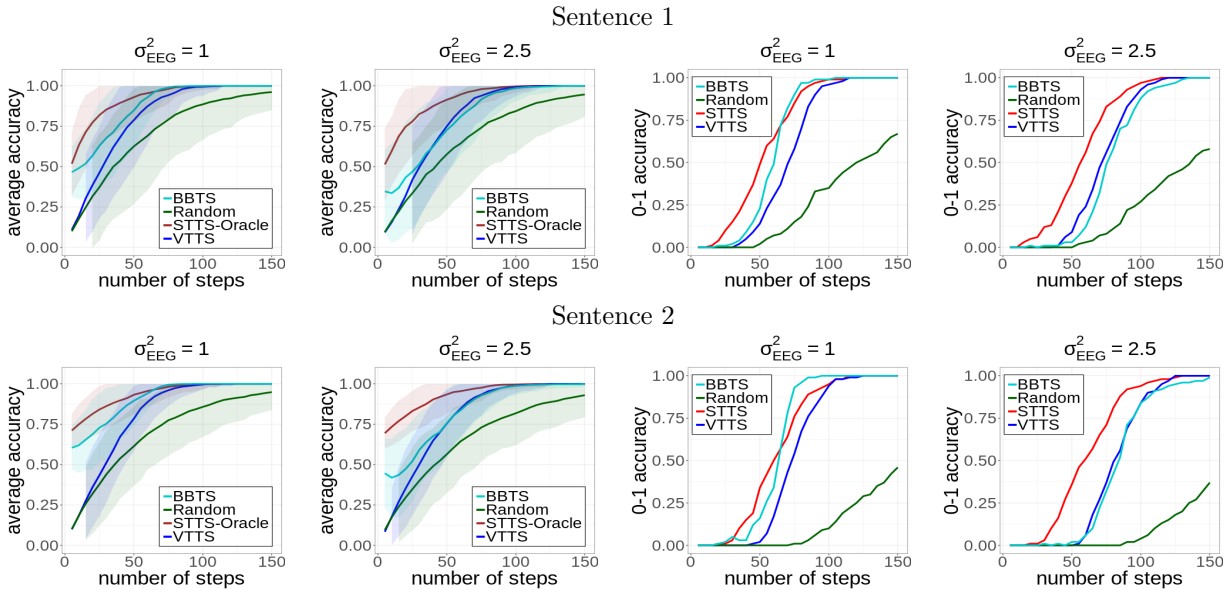

Figure 7: The average accuracy and the 0-1 accuracy for the fixed-budget setting using the P300 speller simulator and the chosen sentences. Two sentences are used, one from a benchmark phrase set, and the other from a news article. A larger $\sigma^2_{\text{EEG}}$ indicates a larger noise level in the simulated EEG signals.

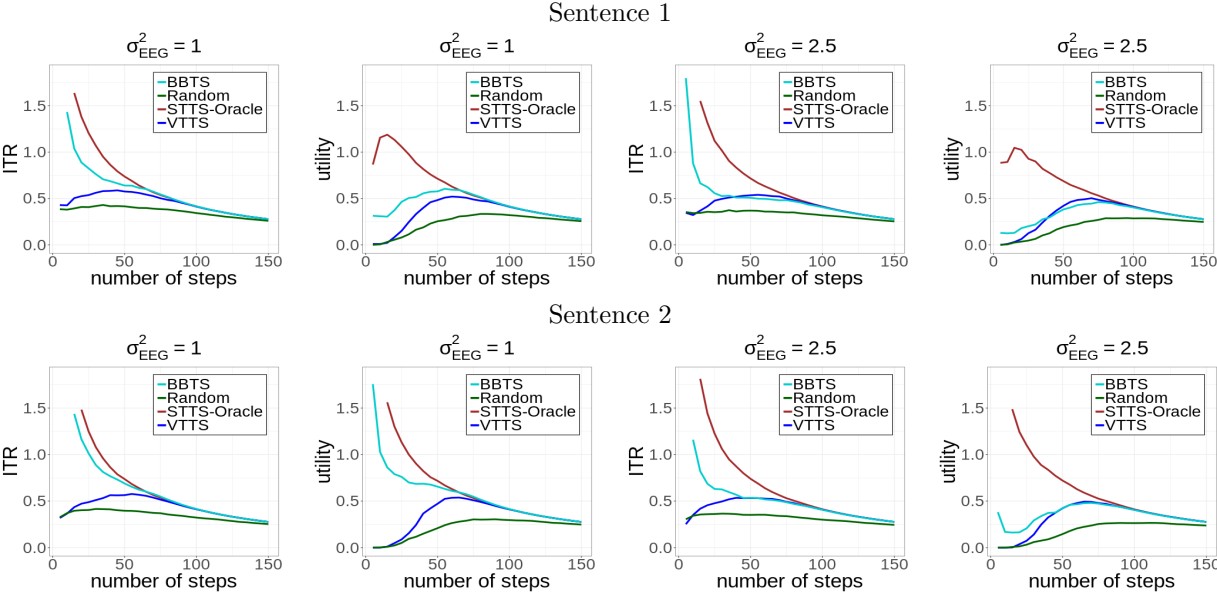

Figure 8: The information transfer rate (ITR) and the utility measure for the fixed-budget setting using the P300 speller simulator and the chosen sentences. Two sentences are used, one from a benchmark phrase set, and the other from a news article. A larger $\sigma^2_{\text{EEG}}$ indicates a larger noise level in the simulated EEG signals.

informed prior, STTS achieves the smallest number of steps, while maintaining about the same accuracy as other methods. It manages to reduce the total number of flashes by 25% to 60%.

For the fixed-budget setting, Figure 7 reports both the 0-1 accuracy and the average accuracy. We see that STTS achieves the highest accuracy compared to the alternative solutions. Figure 8 reports the corresponding computational complexity evaluation measures. Again, STTS outperforms other methods in terms of the ITR and the utility measure.

These results agree with those for the P300 speller simulator example studied in Section 6 and Appendix D with the GPT-generated set of words.

