# OpenReview forum: "Sequential Best-Arm Identification with Application to P300 Speller"
_TMLR — Accepted by TMLR_

### Review · Reviewer_SkCS · 2024-04-18

**Summary Of Contributions:**

The paper studies the use of best arm identification (BAI) by brain-computer interfaces (BCIs), in particular P300 spellers, a type of BCI that selects from a pool of letters/words based on the magnitude of the user's P300 brain responses when each letter/word is highlighted to the user. In this study, each sentence is a task, each word in a sentence is an arm, and the P300 brain responses are the observed rewards. The proposed algorithm uses LLMs as a prior for top-two Thompson sampling. Theoretical results in both the fixed budget setting and the fixed confidence setting to account for the multi-task case is provided. Empirical evaluations of the proposed algorithm are provided in synthetic benchmarks and with a P300 speller simulator built from real world data.

**Audience:**

Yes

**Broader Impact Concerns:**

None.

**Claims And Evidence:**

No

**Requested Changes:**

### Critical to securing recommendation:
1. Please completely specify the synthetic experiments and how the sentences are chosen.
2. In the P300 simulator experiments, either use real sentences used in real world BCI experiments, or justify the choices of prompts and use of GPT-3 generations over real sentences.

### Would strengthen the work:
1. Release code for reproducibility. Ideally, the P300 simulator could be released as well; however, if this is not possible, the code for the synthetic experiments should at least be available.
2. Rewrite the introduction and description of P300 spellers so that it is accessible to non-experts.

**Strengths And Weaknesses:**

### Strengths:
1. The idea of using LLMs as the prior for BCI spellers is interesting and seems to be a useful application of both LLMs and BAI that may have significant real world impact if developed into real products.
2. Theoretical analysis is provided.
3. The proposed algorithm is compared against suitable baselines.
4. There is an attempt to demonstrate real world relevance with the use of the P300 speller simulator (however, please see comments below).

### Weaknesses:
1. **Inadequate description/justification of experiments**. In the synthetic experiments section, it is written "We take one sentence as one experiment, and one word in the sentence as one task, with each sentence consisting of M = 20 words." There is nothing written on where these sentences come from. Are they generated from a LLM as in the P300 simulator experiment, or from some dataset? Lack of this information hinders understanding and reproducibility. In the P300 experiments section, it is written "We use GPT-3 to generate M = 20 words, under a given prompt, as the words that the participant wishes to type, which form the optimal arms. We consider two prompts: Prompt 1 “The most popular food in the United States is”, and Prompt 2 “My favorite sport is”." There is no justification for why these prompts and why LLM generated sentences instead of real sentences used in BCI experiments, of which there should be plenty if there are public BCI data and data upon which the simulator was built. In fact, the use of GPT-3 generated sentences is problematic as GPT-2 is used as the prior, and it is reasonable to expect that GPT-2's training data is at least a subset of that of GPT-3, and therefore that GPT-2 would be a particularly good prior on GPT-3 generations. This does not inspire confidence that the proposed method leads to significant advantage on the distribution of real sentences that BCI users wish to type.
2. **Clarity can be improved**. The writing is unclear in some parts. For example, the description of how the P300 speller works in the introduction was not understandable to me as someone encountering this for the first time. I relied on the description in Sec. 4 of "Brain–Computer Interface Spellers: A Review" by Rezeika et al. (2018) to understand the mechanism.
3. **Lack of reproducibility**. In addition to the inadequate experiment description mentioned in point 1, no code is released. The P300 speller simulator is also unavailable to anyone trying to reproduce the results in this study.
4. **Insignificant contribution to the general BAI literature**. The algorithm is a straightforward application of LLMs as a prior to an existing algorithm, top-two Thompson sampling. The strengths of this paper lie in its application to BCI spellers rather than to the general field.

---

> ### Author Response · Authors · 2024-05-28
> **Point-by-Point Responses to Reviewer SkCS**
>
> We greatly appreciate all your valuable suggestions. We have carefully addressed all the questions, which we believe has led to a much improved paper. In the following, your comments are shown in *italics*, followed by our point-by-point responses. For easy reference, in the revised manuscript, we have colored the text with major changes in red color.
>
> >**Strengths:**
> - *The idea of using LLMs as the prior for BCI spellers is interesting and seems to be a useful application of both LLMs and BAI.*
> - *Theoretical analysis is provided.*
> - *The proposed algorithm is compared against suitable baselines.*
> - *There is an attempt to demonstrate real world relevance with the use of the P300 speller simulator.*
>
> **Response:** Thank you very much for the positive and encouraging feedback!
>
> >**Weaknesses 1:** *Inadequate description/justification of experiments.*
>
> **Response:** We greatly appreciate your questions regarding the text generation. In this revision, we have added more clarification on the setup of P300 speller simulator experiments. In addition, we have considered another scenario to generate the target sentence. That is, following your suggestion, in addition to the text generation using GPT-3, we have added new experiments, by choosing a sentence with $M=9$ words from a benchmark phrase set (MacKenzie and Soukoreff, 2003), and a sentence with $M=15$ words and punctuation marks from a recent news article on *Nature*  (Wong, 2024). We report the new results based on these chosen sentences in Appendix E of the paper.
>
> More specifically, we choose Sentence 1, *"Rent is paid at the beginning of the month"* with $M=9$ words, from a benchmark phrase set (MacKenzie and Soukoreff, 2003). We choose Sentence 2, *"It substantially raises the risk of diseases including cancer, heart disease and diabetes."* with $M=15$ words and punctuation marks, from a recent news article  (Wong, 2024). Note that this news article was published in April, 2024, and thus is not in the GPT-2's training data which is up to October, 2019 only. Similarly as before, we use GPT-2  (Radford et al., 2019) to obtain the prior distribution. However, for the very first word of a sentence, we provide no information at all to GPT-2 for Sentence 1, and we provide the immediately preceding sentence, *"The health harms of smoking tobacco have been established for decades -"* to GPT-2 for Sentence 2. We again conduct experiments for both the fixed-confidence setting and the fixed-budget setting.
>
> For the fixed-confidence setting, Table 3 in Appendix E reports the total number of steps, or flashes, whereas the 0-1 accuracy and the average accuracy for all methods are above 0.9. We see that, facilitated by the LLM-informed prior, STTS achieves the smallest number of steps, while maintaining about the same accuracy as other methods. It manages to reduce the total number of flashes by $25\\%$ to $60\\%$.
>
> For the fixed-budget setting, Figure 7 in Appendix E reports both the 0-1 accuracy and the average accuracy. Figure 8 in Appendix E reports the corresponding computational complexity evaluation measures.
> We see that STTS  achieves the highest accuracy compared to the alternative solutions, and it outperforms other methods in terms of the ITR and the utility measure.
>
> These results agree with those for the P300 speller simulator example studied in Section 6 and Appendix D with the GPT-generated set of words.
>
> Please see Section 6 on Page 10, the third paragraph, for the clarification on the experimental setup. Please see Appendix D on Pages 17 to 19 for the new experimental results with the chosen sentences.
>
> >**Weaknesses 2:** *Clarity can be improved.*
>
> Thank you for the suggestion to improve the clarity. We have updated the paragraph in the introduction regarding the P300 speller.
>
> More specifically, the P300 speller is a primary type of BCI system that allows users to select characters or spell words on a computer screen without using a physical keyboard but instead brain signals. It is based on the P300 event-related potential (ERP), which is a brain response, in the form of a specific pattern of voltage fluctuation, that occurs approximately 300 milliseconds after a relevant target stimulus is presented. Specifically, the system presents an individual character or word through a sequence of flashes, with each flash being a stimulus, usually in a grid-like layout on a virtual screen for the user. If the flash contains the character or word the user wishes to type, an ERP is detected and recorded by a scalp EEG cap or a similar device. In that case, the user elicits a target brain signal. Otherwise, the user elicits a non-target brain signal. The recorded EEG signals are then analyzed by signal processing and machine learning algorithms, which detect the target stimulus and determine the target character. Figure 1 gives a graphical illustration of the P300 speller.
>
> Please see Section 1 on Page 1, the second-to-last paragraph.

---

> ### Author Response · Authors · 2024-05-28
> **Point-by-Point Responses to Reviewer SkCS**
>
> >**Weaknesses 3:** *Lack of reproducibility.*
>
> **Response:** We have released our code for the synthetic experiments at https://anonymous.4open.science/r/P300SequentialBAI-F575 . For the P300 speller simulator, it could be obtained by contacting the authors of Ma et al. (2022).
>
> >**Weaknesses 4:** *Insignificant contribution to the general BAI literature.*
>
> **Response:** We agree that our method essentially integrates the prior information derived from LLMs with the top-two Thompson sampling algorithm. However, we feel it is more than merely a simple extension, and it offers several useful contributions. Notably, our algorithm extends  Qin and Russo (2022); Kocanaogulları et al. (2018) from the setting of a single task to that of multiple and dependent tasks, while relaxing the stringent requirement that all priors have to be correctly specified. Our theory, to the best of our knowledge, is the first work that establishes an explicit error probability bound for top-two Thompson sampling within a fixed-budget context. Finally, as you have pointed out, we apply our method to the P300 speller type BCI experiments, which we hope could pave the way for more adoption of adaptive bandits algorithms in this important application area.
>
> Please see Section 7, Page 11, the last paragraph, and Page 12, the first paragraph.
>
> >**Requested Changes 1:** *(Critical to securing recommendation) Please completely specify the synthetic experiments and how the sentences are chosen.*
>
> **Response:** Done. Please see our detailed response to Weaknesses, \#1.
>
> >**Requested Changes 2:** *(Critical to securing recommendation) Please completely specify the synthetic experiments and how the sentences are chosen.*
>
> **Response:** Done. Please see our detailed response to Weaknesses, \#1.
>
> >**Requested Changes 3:** *(Would strengthen the work) Release code for reproducibility. Ideally, the P300 simulator could be released as well; however, if this is not possible, the code for the synthetic experiments should at least be available.*
>
> **Response:** Done. Please see our detailed response to Weaknesses, \#3.
>
> >**Requested Changes 4:** *(Would strengthen the work) Rewrite the introduction and description of P300 spellers so that it is accessible to non-experts.*
>
> **Response:** Done. Please see our detailed response to Weaknesses, \#2.
>
> **Reference**
>
> Kocanaogulları, A., Marghi, Y. M., Ak¸cakaya, M., and Erdo˘gmu¸s, D. (2018). Optimal query
> selection using multi-armed bandits.
>
> MacKenzie, I. S. and Soukoreff, R. W. (2003). Phrase sets for evaluating text entry techniques.
>
> Qin, C. and Russo, D. (2022). Adaptivity and confounding in multi-armed bandit experiments.
>
> Radford, A., Wu, J., Child, R., Luan, D., Amodei, D., Sutskever, I., et al. (2019). Language models are unsupervised multitask learners.
>
> Wong, C. (2024). Smoking bans are coming: what does the evidence say?

---

### Review · Reviewer_cuzS · 2024-05-07

**Summary Of Contributions:**

This paper proposes a new algorithm for problems inspired by the P300 speller, a brain-computer interface that allows people to "type" words with only their brains.
The authors frame the problem of determining which letter or word is being typed as a multi-armed bandit problem, where the goal is to identify the best arm, which is the letter/arm the person is intending to type.  The key insight is that due to the fact that natural language is being used for communication, existing language models can be used as a prior to help give guidance as to which next letters/words are most likely, given what has already been typed.  The algorithm proposed to address this sequential best-arm identification problem utilizes a top-two Thompson sampling method and can be used either with a confidence or budget termination condition.  An error bound on the algorithm under certain conditions is proven, and the algorithm is experimentally evaluated in a simulated sequential setting, as well as with a P300 simulator, showing in each case improvements over alternative methods.

**Audience:**

Yes

**Claims And Evidence:**

Yes

**Requested Changes:**

- Addition of MAB background and description of how the components of the MAB framework align with the components of the P300 Speller task, as discussed above.  (Critical)
- Some polishing of the notation, at least addressing the questions that I raise above. (Strengthening)

**Strengths And Weaknesses:**

Strengths
- The paper focuses on an important and interesting problem, making a clear contribution over previous methods.
- The combination of multi-armed banded paradigm with the language prior was cleverly done.
- The experimental evaluation of the proposed algorithm was generally well-done and the use of the P300 simulator and the GPT-3 and GPT-2 models was, I thought, a good choice.
- The presence of theoretical analysis of the algorithm is also a strength of the paper.

Weaknesses
- While the text reads clearly and is generally free from typos - there are some higher level things that could be made more clear.  The first time I read the paper entirely through I was still unclear at the end exactly how the multi-armed bandit (MAB) framework was being for the P300 Speller task; what were the MAB actions, what were the rewards, what was the sequence of tasks, etc.  After looking at some of the work that was cited (specifically Heskebeck et al. 2022 was very helpful) I was able to piece together how everything was working and the problem and approach made sense.  I think that this could be made a lot more clear in the paper.  In the beginning of Section 2.1 the variables and components of the problem are defined, but in an abstract way.  This is fine, except for besides connecting the tasks to words, the individual components of the model (actions and reward vector) are not explicitly connected back to the P300 Speller setting.  For me it took a lot of extra effort and reading the other cited work to connect the dots between MAB and the P300 Speller. I would encourage the authors to make these connections more clear so that the motivating problem and surrounding context can be understood from the beginning without prior knowledge of the P300 speller task.
- In a similar vein, I was disappointed in the related work section on multi-armed bandits.  It seemed to omit the basics of what a MAB is and why it is important.  The first sentence of the MAB review starts talking about "best-arm identification in the fixed-confidence setting".  To a reader who isn't the most familiar with MAB these are potentially foreign concepts.  What is the best-arm identification task?  What does fixed confidence mean?  I would encourage the authors to give a short introduction to the basics the MAB setting, describing what is going on, and then carefully define each related task and setting (best-arm identification, fixed-confidence, fixed-budget, batch arm pulls setting, sample-complexity, etc.)  I think this would help the paper be much more comprehensible and impactful to a wider audience.
- Some of the notation got a little bit clunky with the actions being subscripts and having their own subscripts (like on page 5, you have $\mu_{t,a_i,a_j}$)  I wonder if that could be replaced by $\mu_{t,i,j}$ or something, where $i$ and $j$ are then specific actions.  Then on page 8 when describing the Chernoff stopping rule, $\mu_{m,t,j_1}$ is used, which doesn't seem to match up with the previous definition.  Sometimes it seemed like the indices $i$ and $j$ were used to directly refer to actions.  I think this could be adjusted to be more consistent.
Other minor notational issues and questions I had are listed in the following questions:

Questions
- In algorithm 1 on page 5 - the "Pull arm" step is written as if $A_{t,m}$ is a convex combination of $A_{t,m,1}$ and $A_{t,m,2}$, but the text indicates that $A_{t,m}$ should equal either $A_{t,m,1}$ or $A_{t,m,2}$ with $C_{t,m}$ specifying the probability that the first is selected.  I think this should be re-written to be more clear.
- Also in algorithm 1 -when is the mean $\mu_{t,a_i}$ updated, along with the number of pulls $N_{t,a_i}$?  Doesn't this need to happen before the "if max min $Z \geq \gamma$" statement, since $Z$ uses those values?
- On page 5, in the text description of the $Z$ terms,  you say that $KL(\mu_1, \mu_2)$ is the KL-divergence between two distributions with those means.  It seems that the KL divergence cannot be determined from only the means of two distributions, without additional assumptions about those distributions.  Is there something else going on here?
- On page 7 - why does $\mu_{m,t}$ not have a $j$ or $i$ subscript, but $\sigma^2_{m,t,j}$ does?  It seems like these should be indexed the same, as they refer to different parts of the same distribution.
- On page 8 - Do the results for STTS and STTS-Oracle exactly overlap?  As it is, I don't see both results. It looks like only one line is in the figures, but both are included in the figure legend.
- On page 8 - You state that "it is impractical to run the number of steps into the thousands"  How many steps are considered reasonable for this task?  What is the state-of-the-art?

Minor/Typos
- Page 1 "instead the brain signals" could perhaps be "instead brain signals"
- Page 3 "indenpendently" should be "independently"
- Page 4 "We also comment that, SAMP is ..." I think the comma could be omitted.
- Page 14 "recommeded" should be "recommended"

---

> ### Author Response · Authors · 2024-05-28
> **Point-by-Point Responses to Reviewer cuzS**
>
> We greatly appreciate all your valuable suggestions. We have carefully addressed all the questions, which we believe has led to a much improved paper. In the following, your comments are shown in *italics*, followed by our point-by-point responses. For easy reference, in the revised manuscript, we have colored the text with major changes in red color.
>
> >**Strengths:**
> - *The paper focuses on an important and interesting problem, making a clear contribution over previous methods.*
> - *The combination of multi-armed banded paradigm with the language prior was cleverly done.*
> - *The experimental evaluation of the proposed algorithm was generally well-done and the use of the P300 simulator and the GPT-3 and GPT-2 models was, I thought, a good choice.*
> - *The presence of theoretical analysis of the algorithm is also a strength of the paper.*
>
> **Response:**  Thank you very much for the positive and encouraging feedback!
>
> >**Weaknesses 1:** *I would encourage the authors to make these connections more clear so that the motivating problem and surrounding context can be understood from the beginning without prior knowledge of the P300 speller task.*
>
> **Response:** We greatly appreciate this suggestion. In this revision, we have added more clarification on how a P300 speller experiment can be cast in our problem formulation.
>
> More specifically, in the context of a P300 speller experiment, take the identification of a set of words as an example. An agent refers to the system (not the user). A task means identifying a single word the user intends to type, which forms a single bandit environment. An action, or say, pulling an arm, means flashing one word on the virtual user screen, and the reward is the classifier score computed based on the captured EEG signals and a pre-trained binary classifier. A decision is the word the system believes the user intends to type. A budget is the total number of flashes the system is to make for a single task. In this setup, $M$ is the total number of words the user intends to type, and $J$ is the total number of flashes.
>
> Please see Section 2.1 on Page 4, the second paragraph.
>
> >**Weaknesses 2:** *I would encourage the authors to give a short introduction to the basics the MAB setting, describing what is going on, and then carefully define each related task and setting.*
>
> **Response:** Thank you again for this suggestion. In this revision, we have added more background introduction for the multi-armed bandit problem. We have also updated the literature review on the single task learning to provide more explanation of some key terminology.
>
> Specifically, the multi-armed bandit (MAB) problem  (Robbins, 1952) is a classic scenario in decision theory and reinforcement learning that studies the problem of balancing exploration and exploitation. It involves an agent that seeks to optimize actions that maximize expected rewards. The agent must explore the action space sufficiently to acquire the knowledge needed to exploit the best action. Best-arm identification (BAI) is a variant of MAB, where the learner's objective is to identify the optimal arm, i.e., the arm with the highest expected reward, with a high accuracy. BAI is especially relevant to the P300 speller problem, where the goal is to swiftly and accurately identify target characters or words.
>
> For learning a single task, Even-Dar et al. (2002) first introduced best-arm identification
> in the fixed-confidence setting, ensuring a specified confidence level in the correctness of
> the identification. Audibert et al. (2010) studied the fixed-budget setting, where the goal
> is to identify the best arm within a given number of trials or budget. In the BAI problem,
> we use sample complexity, the number of samples or pulls needed to correctly identify
> the best arm, to measure the theoretical properties of the algorithms. Kaufmann et al.
> (2016) investigated optimal sample complexity, and Russo (2016) proposed the top-two
> Thompson sampling as an effective anytime sampling rule that does not depend on the
> confidence parameter. Its theoretical properties were studied in Russo (2016); Qin et al.
> (2017); Shang et al. (2020); Qin and Russo (2022); Jourdan et al. (2022). However, most
> existing asymptotic analyses did not quantify the prior effect. An exception is Qin and
> Russo (2022), who showed the prior effect through entropy in the simple regret setting.
> Jun et al. (2016) explored scenarios where multiple pulls of the arms can be conducted
> simultaneously or in batches, rather than sequentially.
>
> Please see Section 1 on Page 2, the last paragraph, and Page 3, the first paragraph.

---

> ### Author Response · Authors · 2024-05-28
> **Point-by-Point Responses to Reviewer cuzS**
>
> >**Weaknesses 3:** *Some of the notation got a little bit clunky with the actions being subscripts and having their own subscripts.*
>
> **Response:** Thank you for pointing this out! Following your suggestions, we now use $N_{t,i}$ to denote the number of pulls of arm $a_i$ before step $t$ for each task, use $\mu_{t, i}$ to denote the posterior mean for arm $a_i$, and let $\mu_{t, i, j} = (N_{t, i}\mu_{t, i}+N_{t, j}\mu_{t, j})/(N_{t, i}+N_{t, j})$.
>
> Please see Section 3.2 on Page 5, Lines 16 to 17.
>
> >**Question 1:** *In algorithm 1 on page 5 - the "Pull arm" step is written as if $A_{t,m}$ is a convex combination of $A_{t,m,1}$ and $A_{t,m,2}$, but the text indicates that $A_{t,m}$ should equal either $A_{t,m,1}$ or $A_{t,m,2}$ with $C_{t,m}$ specifying the probability that the first is selected.*
>
> **Response:** We appreciate this question. We have updated our Algorithm 1 to discuss how to update $A_{t,m}$. Please see Section 3.2, Algorithm 1, on Page 6.
>
> >**Question 2:** *Also in algorithm 1 -when is the mean $\mu_{t,a_i}$ updated, along with the number of pulls $N_{t,a_i}$? Doesn't this need to happen before the "if max min $Z \geq \gamma$" statement, since $Z$ uses those values?*
>
> **Response:** We appreciate this question. We have updated our Algorithm 1 to discuss when to update those parameters.  Please see Section 3.2, Algorithm 1, on Page 6.
>
> >**Question 3:** *On page 5, in the text description of the $Z$ terms, you say that $KL(\mu_1, \mu_2)$ is the KL-divergence between two distributions with those means. It seems that the KL divergence cannot be determined from only the means of two distributions, without additional assumptions about those distributions. *
>
> **Response:** Thanks for pointing this out! Here we follow Garivier and Kaufmann (2016) and focus on the simple case where the distributions are fully parameterized by their means, and we use $\text{KL}(\mu_1, \mu_2)$ to denote the KL-divergence between the two distributions with means $\mu_1$ and $\mu_2$, respectively. Please see Section 3.2 on Page 5, Lines 19 to 21.
>
> >**Question 4:** *On page 7 - why does $\mu_{m,t}$ not have a $j$ or $i$ subscript, but $\sigma^2_{m,t,j}$ does? It seems like these should be indexed the same, as they refer to different parts of the same distribution.*
>
> **Response:**  Sorry, it was a typo. Now we have updated the notation to $\mu_{m,t.j}$. Please see Section 3.2 on Page 8, Line 11.
>
> >**Question 5:** *On page 8 - Do the results for STTS and STTS-Oracle exactly overlap? As it is, I don't see both results. It looks like only one line is in the figures, but both are included in the figure legend.*
>
> **Response:** Yes, these two results overlap, indicating that STTS and STTS-Oracle  exhibit similar performances. This is because, as the accuracy of STTS  is consistently close to one, $P(A_m^* = . |A^*_{m-1} = a_{m-1}^*)$,
> and
> $P(A_m^* = . |A^*_{m-1} = \psi_{m-1})$ are very close, and thus the prior information from STTS  and STTS-Oracle  are almost the same. Please see Section 5.2 on Page 8, Lines -6 to -4.
>
> >**Question 6:** *On page 8 - You state that "it is impractical to run the number of steps into the thousands" How many steps are considered reasonable for this task? What is the state-of-the-art?*
>
> **Response:** In most current P300 speller experiments, the random policy is used, and it usually requires 100 to 150 steps (flashes) for each task  (Kindermans et al., 2012). In our P300 speller simulator example, the random policy takes about 150 steps to reach the 95\% accuracy, which agrees with the practical observations. Please see Section 6 on Page 10, Line -1, and Page 11, Lines 1 to 2.
>
> >**Minor 1:** *Page 1 "instead the brain signals" could perhaps be "instead brain signals"*
>
> **Response:** Corrected.  Please see Page 1, Line -13.
>
> >**Minor 2:** *Page 3 "indenpendently" should be "independently"*
>
> **Response:** Corrected. Please see Page 3, Line 16.
>
> >**Minor 3:** *Page 4 "We also comment that, SAMP is ..." I think the comma could be omitted."*
>
> **Response:** Corrected. Please see Page 5, Line 10.
>
> >**Minor 4:** *Page 14 "recommeded" should be "recommended".*
>
> **Response:** Corrected. Please see Page 14, Line -14.
>
> >**Requested Change 1:** *Addition of MAB background and description of how the components of the MAB framework align with the components of the P300 Speller task, as discussed above. (Critical)*
>
> **Response:** Done. Please see our detailed responses to Weaknesses, \#1 and \#2.
>
> >**Requested Change 2:** *Some polishing of the notation, at least addressing the questions that I raise above. (Strengthening)*
>
> **Response:** Done. Please see our detailed response to Weaknesses, \#3.
>
> **Reference**
>
> Garivier, A. and Kaufmann, E. (2016). Optimal best arm identification with fixed confidence.
>
> Kindermans, P.-J., Verstraeten, D., and Schrauwen, B. (2012). A bayesian model for exploiting application constraints to enable unsupervised training of a p300-based bci.

---

### Review · Reviewer_5hRY · 2024-05-14

**Summary Of Contributions:**

This paper proposes to use bandit algorithms to improve the
performance of a brain-computer interface system, called "P300
speller". The proposed algorithm tries to predict what will be the
next word that the user wants to output.

**Audience:**

Yes

**Broader Impact Concerns:**

There is no impact statement.

**Claims And Evidence:**

No

**Requested Changes:**

The model should be clarified and its relationship with the title and introduction should be precised.

**Strengths And Weaknesses:**

# Strengths
- The problem seems motivated by a concrete application that is
  relevant.
- Numerical results seem to indicate that the solution performs well.

# Weaknesses
The theoretical formulation of the problem is not clear to me: I did
not understand the relationship between the exposed problem in section
2 and the application to the P300 speller. For instance:
- what do M, m and j correspond to?
- what is an arm?
- why is the noise Gaussian?
- what is the feedback? and who provides the feedback?
- why is the reward as specified in the last equation of section 2.2?
- It seems to me that the prior plays an important role... But what is
  the signal that we observe before taking a decision (EEG or EDCoG) ?
  Why is it not in the model?

---

> ### Author Response · Authors · 2024-05-28
> **Point-by-Point Responses to Reviewer 5hRY**
>
> We greatly appreciate all your valuable suggestions. We have carefully addressed all the questions, which we believe has led to a much improved paper. In the following, your comments are shown in *italics*, followed by our point-by-point responses. For easy reference, in the revised manuscript, we have colored the text with major changes in red color.
>
> >**Strengths:**
> - *The problem seems motivated by a concrete application that is relevant.*
> - *Numerical results seem to indicate that the solution performs well.*
>
> **Response:** Thank you very much for the positive and encouraging feedback!
>
> >**Weaknesses 1:** *The theoretical formulation of the problem is not clear to me: I did not understand the relationship between the exposed problem in section 2 and the application to the P300 speller.*
>
> **Response:** We greatly appreciate this suggestion. In this revision, we have added more clarification on how a P300 speller experiment can be cast in our problem formulation.
>
> More specifically, in the context of a P300 speller experiment, take the identification of a set of words as an example. An agent refers to the system (not the user). A task means identifying a single word the user intends to type, which forms a single bandit environment. An action, or say, pulling an arm, means flashing one word on the virtual user screen, and the reward is the classifier score computed based on the captured EEG signals and a pre-trained binary classifier. A decision is the word the system believes the user intends to type. A budget is the total number of flashes the system is to make for a single task. In this setup, $M$ is the total number of words the user intends to type, and $J$ is the total number of flashes.
>
> Please see Section 2.1 on Page 4, the second paragraph.
>
> >**Weaknesses 2:** *What do $M$, $m$ and $j$ correspond to?*
>
> **Response:**
> In the P300 speller experiment, $M$ is the total number of words the user intends to type, and $m$ indicates the $m$th word in a sentence or a set of words, $m = 1, \ldots, M$. $J$ is the total number of flashes (actions) for each word shown on the virtual screen, and $j$ indicates the $j$th flash (action), $j=1, \ldots, J$.
>
> Please see Section 2.1 on Page 4, Lines 9 to 10.
>
> >**Weaknesses 3:** *What is an arm?*
>
> **Response:**
> An arm means flashing one word on the virtual screen.
>
> >**Weaknesses 4:** *Why is the noise Gaussian?*
>
> **Response:**
> The Gaussian noise is commonly used in the P300 setup, for example, Ma et al. (2021). So we employ the Gaussian noise in our simulations too.
>
> >**Weaknesses 5:** *What is the feedback? And who provides the feedback?*
>
> **Response:**
> The feedback essentially means the reward in the P300 speller experiment. After one flash (an action), the EEG signals are collected, and the reward is the classifier score computed based on the captured EEG signals and a pre-trained binary classifier. The system does not require any human feedback here.
>
> >**Weaknesses 6:** *Why is the reward as specified in the last equation of section 2.2?*
>
> **Response:**
> Here we follow the same form of the reward as in Ma et al. (2021). Basically, the system presents an individual character or word through a sequence of flashes, with each flash being a stimulus, usually in a grid-like layout on a virtual screen for the user. If the flash contains the character or word the user wishes to type, an ERP is detected and recorded by a scalp EEG cap or a similar device. In that case, the user elicits a target brain signal. Otherwise, the user elicits a non-target brain signal. The recorded EEG signals are then analyzed by signal processing and machine learning algorithms, which detect the target stimulus and determine the target character based on the reward.
>
> >**Weaknesses 7:** *It seems to me that the prior plays an important role... But what is the signal that we observe before taking a decision (EEG or EDCoG) ? Why is it not in the model?*
>
> **Response:**
> The signals we observe in a P300 speller experiment are only the EEG signals, based on which the system seeks to identify the character or the word the user intends to type.

---

> > ### Author Response · Authors · 2024-05-28
> > **Point-by-Point Responses to Reviewer 5hRY**
> >
> > >**Requested changes:** *The model should be clarified and its relationship with the title and introduction should be precised.*
> >
> > **Response:** We appreciate all your suggestions regarding the clarity of our presentation. We have added more clarifications about the P300 speller and how it is connected with our sequential multi-armed bandit formulation.
> >
> > Specifically, the P300 speller is a primary type of BCI system that allows users to select characters or spell words on a computer screen without using a physical keyboard but instead brain signals. It is based on the P300 event-related potential (ERP), which is a brain response, in the form of a specific pattern of voltage fluctuation, that occurs approximately 300 milliseconds after a relevant target stimulus is presented. Specifically, the system presents an individual character or word through a sequence of flashes, with each flash being a stimulus, usually in a grid-like layout on a virtual screen for the user. If the flash contains the character or word the user wishes to type, an ERP is detected and recorded by a scalp EEG cap or a similar device. In that case, the user elicits a target brain signal. Otherwise, the user elicits a non-target brain signal. The recorded EEG signals are then analyzed by signal processing and machine learning algorithms, which detect the target stimulus and determine the target character. Figure 1 gives a graphical illustration of the P300 speller.
> >
> > To formulate a P300 speller experiment as a multi-armed bandit problem, take the identification of a set of words as an example. An agent refers to the system (not the user). A task means identifying a single word the user intends to type, which forms a single bandit environment. An action, or say, pulling an arm, means flashing one word on the virtual user screen, and the reward is the classifier score computed based on the captured EEG signals and a pre-trained binary classifier. A decision is the word the system believes the user intends to type. A budget is the total number of flashes the system is to make for a single task. In this setup, $M$ is the total number of words the user intends to type, and $J$ is the total number of flashes.
> >
> > Please see Section 1 on Page 1, the second-to-last paragraph, and Section 2.1 on Page 4, the second paragraph.
> >
> > **Reference**
> >
> > Ma, T., Huggins, J. E., and Kang, J. (2021). Adaptive sequence-based stimulus selection in an erp-based brain-computer interface by thompson sampling in a multi-armed bandit problem.

---

### Author Response · Authors · 2024-05-28
**Summary of Revision**

We thank the Editor, the Associate Editor, and the three reviewers for all your valuable suggestions. We have carefully addressed all the questions. Here is a summary of our major changes, followed by our point-by-point responses to each reviewer.
- We have added new numerical experiments for the P300 speller example, by choosing a
sentence from a benchmark phrase set and a sentence from a recent news article.
- We have added more clarifications about the P300 speller, and how it is connected with
our sequential multi-armed bandit formulation.
- We have updated the literature review on the multi-armed bandit problem, and added a
brief discussion to clarify our contributions.
- We have released our code online.
- We have updated numerous notations to make them more consistent throughout the
article.

We attach a **full rebuttal letter** in the front of the revised manuscript (please download the revised paper) and also respond point by point in openreview. The change in the revised manuscript is highlighted in red.

---

### Decision · Action_Editor_7TLZ · 2024-07-21

**Recommendation:** Accept with minor revision

**Comment:**

There were concerns on the presentation, which were raised by most of reviewers. While the paper is now in a better shape after revision, one reviewer is not satisfied yet, claiming that he/she does not see clearly the relationship between the theoretical model and the practical problem to solve and the model should be better explained. The same thing can happen for many readers who wish to see a real-world application of the best-arm identification but are not familiar with BCI and P300 speller. Thus I would like to suggest the authors to elaborate Fig. 1 such that
it clearly illustrates how a sequence of best-arm identification is formulated in the context of a P300 speller. Once what is going on becomes clear in the first place, the rest of things will be easily caught up.

**Audience:**

This paper demonstrates an interesting application of theoretical best-arm identification technique. Thus, it will be interesting for readers who actually wish to see  how the best-arm identification is used to solve a real-world problem.

**Claims And Evidence:**

This paper casts one of widely-studied BCI problems, P300 speller as a sequence of best-arm identification tasks within the context of multi-armed bandits. The key contribution in this paper may lie in the idea of leveraging language models as a prior to help to give guidance for which next words are most likely. It is claimed that a sequential top-two Thomson sampling algorithm is proposed. However, the top-two algorithm has been widely used for the best-arm identification, so the corresponding contribution is not big. Experiments demonstrate the empirical improvement through both simulations as well as the data involving the real BCI experiments.

---

> ### Author Response · Authors · 2024-08-18
> **Responses to Action Editor 7TLZ**
>
> Thank you for tentatively accepting our paper, and for your excellent suggestion. In this revision, we have updated the caption of the illustrative Figure 1, by adding more details on how the P300 speller experiment can be formulated as a series of best-arm identification problems.